# Generalization Error Bounds for Graph Embedding Using Negative Sampling: Linear vs Hyperbolic

**Atsushi Suzuki**
University of Greenwich
London, United Kingdom
atsushi.suzuki.rd@gmail.com

**Atsushi Nitanda**
Kyushu Institute of Technology
Fukuoka, Japan
nitanda@ai.kyutech.ac.jp

**Jing Wang**\*
University of Greenwich
London, United Kingdom
jing.wang@greenwich.ac.uk

**Linchuan Xu**
The Hong Kong Polytechnic University
Hong Kong SAR, China
linch.xu@polyu.edu.hk

**Kenji Yamanishi**
The University of Tokyo
Tokyo, Japan
yamanishi@g.ecc.u-tokyo.ac.jp

**Marc Cavazza**
National Institute of Informatics
Tokyo, Japan
marc.cavazza@gmail.com

## Abstract

Graph embedding, which represents real-world entities in a mathematical space, has enabled numerous applications such as analyzing natural languages, social networks, biochemical networks, and knowledge bases. It has been experimentally shown that graph embedding in hyperbolic space can represent hierarchical tree-like data more effectively than embedding in linear space, owing to hyperbolic space's exponential growth property. However, since the theoretical comparison has been limited to ideal noiseless settings, the potential for the hyperbolic space's property to worsen the generalization error for practical data has not been analyzed. In this paper, we provide a generalization error bound applicable for graph embedding both in linear and hyperbolic spaces under various negative sampling settings that appear in graph embedding. Our bound states that error is polynomial and exponential with respect to the embedding space's radius in linear and hyperbolic spaces, respectively, which implies that hyperbolic space's exponential growth property worsens the error. Using our bound, we clarify the data size condition on which graph embedding in hyperbolic space can represent a tree better than in Euclidean space by discussing the bias-variance trade-off. Our bound also shows that imbalanced data distribution, which often appears in graph embedding, can worsen the error.

## 1 Introduction

Graphs are a fundamental formulation of real-world entities and their relations, such as words in natural languages, people in social network, and objects in knowledge bases. Here, the vertices and edges of a graph correspond to the entities and the relations among them, respectively. Based on the formulation, graph embedding has enabled numerous applications for those data, such as machine translation and sentiment analysis for natural language [1, 2, 3, 4], and community detection and

---

\*Corresponding author

35th Conference on Neural Information Processing Systems (NeurIPS 2021).

link prediction for social network data [5, 6, 7, 8, 9], pathway prediction of biochemical network [10, 11], and link prediction and triplet classification for knowledge base [12, 13, 14, 15, 16, 17]. Graph embedding produces representations of a graph's vertices in a space equipped with a function that defines the dissimilarity between two points. In this paper, we call a function that defines the dissimilarity a *dissimilarity function* and a space equipped with a dissimilarity function a *dissimilarity space*. For example, we can consider the squared distance as a dissimilarity function. Graph embedding aims to obtain representations such that the dissimilarity function reflects the relations defined by the edges. Specifically, we expect that the dissimilarity function returns a small value for the representations of a *positive pair*, a pair of vertices connected by an edge, and a large value for a *negative pair*, a pair not connected. Here, to reduce the computational cost, obtaining training data by the *negative sampling* strategy has been known to be effective [1]. In this strategy, we sample a positive pair for each iteration, followed by sampling negative pairs around the positive pair.

As a dissimilarity space, many graph embedding methods have used linear space equipped with an inner product function [1, 2, 3, 5, 6, 7, 8, 9], which we call *linear graph embedding (LGE)*. However, linear space has limitations in representing data with a hierarchical tree-like structure [18, 19, 20, 21]. These limitations are due to linear space's polynomial growth property, which means that the volume or surface of a ball in linear space grows polynomially with respect to its radius. This linear space's growth speed is significantly slower than embedding hierarchical data such as an $r$-ary tree ($r \geq 2$) requires, which is exponential [21]. To overcome this limitation, graph embedding in hyperbolic space has recently attracted much attention [20, 22, 23, 24, 25, 26, 4, 27], which we call *hyperbolic graph embedding (HGE)* in this paper. In contrast to linear space's polynomial growth property, hyperbolic space has the exponential growth property, that is, the volume of any ball in hyperbolic space grows exponentially with respect to its radius [18, 19, 20, 21]. As a result, hyperbolic space is almost tree-like in that it can be well approximated by a tree [28], and we can embed any tree in hyperbolic space with arbitrarily low distortion [29]. Existing HGE papers have experimentally shown HGE's ability to effectively represent hierarchical tree-like data such as taxonomies and social networks. However, the theoretical guarantee of HGE's performance is limited to ideal noiseless settings [28, 29, 23], and the comparison between LGE and HGE's generalization performance in noisy settings has not been discussed, although HGE could have a much worse generalization error than LGE in compensation for hyperbolic space's exponential growth property and cause overfitting for real data, which are often noisy.

In this paper, we derive a generalization bound for graph embedding using the negative sampling strategy under noisy settings. To the best of our knowledge, this is the first work that derives a complete generalization error bound for both LGE and HGE. As discussed in [21], since the generalization error of a learning model reflects the volume of its hypothesis space, we can conjecture that the generalization errors of LGE and HGE are polynomial and exponential with respect to the embedding space's radius, reflecting inner product space and hyperbolic space's polynomial and exponential growth property. Also, an imbalanced data distribution, which often appears in graph embedding reflecting the graph structure's imbalance, may worsen the error. In this paper, we formally prove that the above conjectures are true, as well as clarify the dependency of the error bound on the number of entities and the size of training data. Based on the derived generalization error bounds, we also clarify the data size condition on which HGE outperforms LGE in embedding a tree, by discussing the bias-variance trade-off.

To derive a generalization error bound in embedding problem, existing papers [30, 21] deriving ordinal embedding's bounds have converted the problem into a linear prediction problem to calculate its Rademacher complexity [31, 32, 33]. Also, for hyperbolic embedding, the decomposition of the Lorentz Gramian matrix [34] has been combined with the above technique [21]. For graph embedding using negative sampling, however, we cannot straightforwardly apply these techniques, which have been effective for deriving ordinal embedding's generalization error bound. Since the data distribution depends on the graph structure and positive sampling affects the distribution of negative sampling in the negative sampling structure, we cannot apply the i.i.d.-uniform-distribution-based discussions in [30, 21] to graph embedding using the negative sampling strategy. Although a recent unpublished paper [35] has attempted to derive a generalization error bound only for LGE, which is incomplete in that the bound still has an unevaluated part and cannot be applied for noisy settings, this dependency between the distributions has been ignored. We solve this problem by decomposing the loss function into functions of each edge sampled by the negative sampling strategy. We achieved this decomposition by our novel multivariable version of the Ledoux-Talagrand contraction lemma

[36]. By our approach, we can upper-bound the Rademacher complexity of a graph embedding loss function by the sum of the Rademacher complexities [31, 32, 33] of linear prediction models, which have been calculated in [30, 21]. As a result, our generalization bound is valid for various negative sampling settings where the distribution of positive and negative edges are dependent.

Our contributions are threefold:

- We have derived the generalization error bound for negative-sampling-based graph embedding. Our theorem is applicable to various settings regarding the embedding space, data distribution and loss function, such as LGE and HGE, the dependency between the positive pairs and negative pairs' distribution, and sigmoid loss functions. Our upper bound shows that LGE and HGE cause a polynomial and exponential error with respect to the embedding space's radius, respectively. Our bound also shows that imbalanced data distribution can worsen the error.

- We have derived specific error bounds for practical negative sampling strategies.

- We have derived an explicit training data size condition on which HGE can represent a tree better than LGE.

## 2 Preliminary

**Notation** In this paper, the symbol $:=$ is used to state that its left hand side is defined by its right hand side. We denote by $\mathbb{Z}, \mathbb{Z}_{>0}, \mathbb{R}, \mathbb{R}_{\geq 0}$ the set of integers, the set of positive integers, the set of real numbers, and the set of non-negative real numbers, respectively. Suppose that $D, V \in \mathbb{Z}_{>0}$. We denote by $, \mathbb{R}^D$ the set of $D$-dimensional real vectors. For a matrix $\boldsymbol{A} \in \mathbb{R}^{D,V}$, we denote by $[\boldsymbol{A}]_{d,v}$ the element in the $d$-row and the $v$-th column. For a vector $\boldsymbol{x} \in \mathbb{R}^D$, we denote by $\|\boldsymbol{x}\|_2$ the 2-norm of $\boldsymbol{x}$, defined by $\|\boldsymbol{x}\|_2 = \sqrt{\boldsymbol{x}^\top \boldsymbol{x}}$, and for a matrix $\boldsymbol{A} \in \mathbb{R}^{D,V}$, we denote by $\|\boldsymbol{A}\|_{\mathrm{op},2}$, the operator norm of $\boldsymbol{A}$ with respect to the 2-norm, defined by $\|\boldsymbol{A}\|_{\mathrm{op},2} := \sup_{\boldsymbol{x} \in V, \boldsymbol{x} \neq \boldsymbol{0}} \frac{\|\boldsymbol{A}\boldsymbol{x}\|_2}{\|\boldsymbol{x}\|_2}$. By $(a_n)_{n=1}^N$, we denote a sequence $(a_1, a_2, \ldots, a_N)$.

### 2.1 Graph Embedding

In this section, we first formulate the general embedding problem, before we specialize it into a graph embedding. Consider an entity set $\mathcal{V}$ and the *true dissimilarity function* $\mathcal{V} \times \mathcal{V} \to \mathbb{R}$ defined on the entity set $\mathcal{V}$. For $i, j \in \mathcal{V}$, we call $\delta^*(i, j)$ the *true dissimilarity* between $i$ and $j$, and a small $\delta^*(i, j)$ value implies that entity $i$ and $j$ are similar or closely related to each other, and large $\delta^*(i, j)$ value implies its converse. In this paper, we identify $\mathcal{V}$ with the integer set $\{1, 2, \ldots, |\mathcal{V}|\}$. Embedding aims to get representations $z_1, z_2, \ldots, z_V$ of the entity set $\mathcal{V}$ in a space $(\mathcal{Z}, \delta_{\mathcal{Z}})$ equipped with a dissimilarity function $\delta_{\mathcal{Z}} : \mathcal{Z} \times \mathcal{Z} \to \mathbb{R}$ of $\mathcal{Z}$ so that the representations are consistent to the true dissimilarity among the entities in that if $\delta^*(i, j)$ is small, then $\delta(z_i, z_j)$ is also small, and vice versa. We call $\mathcal{Z}$ the *embedding space*. Specifically, we aim to satisfy

$$\delta^*(i, j) \lesseqgtr \theta^* \Leftrightarrow \delta_{\mathcal{Z}}(z_i, z_j) \lesseqgtr \theta_{\mathcal{Z}}, \tag{1}$$

as frequent as possible with respect to some distribution regarding $(i, j)$, which we discuss in the next subsection. Here, $\theta^*, \theta_{\mathcal{Z}} \in \mathbb{R}_{\mathcal{Z}}$ are fixed thresholds regarding the true dissimilarity $\delta^*$ and the dissimilarity $\delta_{\mathcal{Z}}$ in the embedding space.

As a dissimilarity function of the embedding space $\mathcal{Z}$, we mainly consider the square distance $[\Delta_{\mathcal{Z}}(z_i, z_j)]^2$ if $\mathcal{Z}$ is equipped with a distance function $\Delta_{\mathcal{Z}} : \mathcal{Z} \times \mathcal{Z} \to \mathbb{R}_{\geq 0}$, and the negative inner product $-\langle z_i, z_j \rangle$ if $\mathcal{Z}$ is equipped with an inner product $\langle \cdot, \cdot \rangle : \mathcal{Z} \times \mathcal{Z} \to \mathbb{R}$. In this paper, we deal with the following four dissimilarity spaces.

**Definition 1.** (Dissimilarity spaces)

**(a)** The $D$-dimensional Euclidean space $(\mathbb{R}^D, \Delta_{\mathbb{R}^D})$ consists of the set of $D$-dimensional real vectors and the distance function $\Delta_{\mathbb{R}^D} : \mathbb{R}^D \times \mathbb{R}^D \to \mathbb{R}_{\geq 0}$ defined by $\Delta_{\mathbb{R}^D}(\boldsymbol{z}, \boldsymbol{z}') := \|\boldsymbol{z} - \boldsymbol{z}'\|_2^2$. The dissimilarity function is given by $\delta_{\mathbb{R}^D}(\boldsymbol{z}, \boldsymbol{z}') := [\Delta_{\mathbb{R}^D}(\boldsymbol{z}, \boldsymbol{z}')]^2$.

**(b)** The $D$-dimensional hyperbolic space $\left(\mathbb{L}^D, \Delta_{\mathbb{L}^D}\right)$ consists of the $D$-dimensional hyperboloid $\mathbb{L}^D \subset \mathbb{R}^{D+1}$ and the distance function $\Delta_{\mathbb{L}^D}$ defined by

$$\mathbb{L}^D := \left\{ \boldsymbol{x} \in \mathbb{R}^{D+1} \mid \langle \boldsymbol{x}, \boldsymbol{x} \rangle_{\mathrm{M}} = -1 \right\}, \qquad \Delta_{\mathbb{L}^D}(\boldsymbol{x}, \boldsymbol{x}') := \mathrm{arcosh}(-\langle \boldsymbol{x}, \boldsymbol{x}' \rangle_{\mathrm{M}}), \qquad (2)$$

where $\langle \cdot, \cdot \rangle_{\mathrm{M}} : \mathbb{R}^D \times \mathbb{R}^D \rightarrow \mathbb{R}$ is the Minkowski inner prodct defined by $\left\langle \begin{bmatrix} z_0 & z_1 & \cdots & z_D \end{bmatrix}^\top, \begin{bmatrix} z'_0 & z'_1 & \cdots & z'_D \end{bmatrix}^\top \right\rangle_{\mathrm{M}} := -z^0 z'^0 + \sum_{d=1}^{D} z_d z'_d$. The dissimilarity function is given by $\delta_{\mathbb{L}^D}(\boldsymbol{z}, \boldsymbol{z}') := \left[\Delta_{\mathbb{L}^D}(\boldsymbol{z}, \boldsymbol{z}')\right]^2$.

**(c)** The $D$-dimensional sphere $\left(\mathbb{S}^D, \Delta_{\mathbb{S}^D}\right)$ consists of the subset $\mathbb{S}^D \subset \mathbb{R}^{D+1}$ and the distance function $\Delta_{\mathbb{L}^D}$ defined by

$$\mathbb{S}^D := \left\{ \boldsymbol{x} \in \mathbb{R}^{D+1} \mid \boldsymbol{x}^\top \boldsymbol{x} = 1 \right\}, \qquad \Delta_{\mathbb{S}^D}(\boldsymbol{x}, \boldsymbol{x}') := \arccos(\boldsymbol{x}^\top \boldsymbol{x}'). \qquad (3)$$

The dissimilarity function is given by $\delta_{\mathbb{S}^D}(\boldsymbol{z}, \boldsymbol{z}') := \left[\Delta_{\mathbb{S}^D}(\boldsymbol{z}, \boldsymbol{z}')\right]^2$.

**(d)** The canonical $D$-dimensional inner product space $\left(\mathbb{I}^D, \delta_{\mathbb{I}^D}\right)$ as a dissimilarity space consists of the set $\mathbb{I}^D = \mathbb{R}^D$ of $D$-dimensional real vectors and the dissimilarity function $\delta_{\mathbb{I}^D} : \mathbb{I}^D \times \mathbb{I}^D \rightarrow \mathbb{R}$ defined by the negative canonical inner product $\delta_{\mathbb{I}^D}(\boldsymbol{z}, \boldsymbol{z}') := -\boldsymbol{z}^\top \boldsymbol{z}'$.

Our main focus in this paper is $\mathbb{R}^D$ and $\mathbb{L}^D$, although our bound is also applicable to $\mathbb{S}^D$ and $\mathbb{I}^D$.

**Remark 1.** There are multiple models to represent the above spaces. For example, we can use the Poincaré ball model, upper half space model, Klein ball model to represent hyperbolic space, other than the hyperboloid model used in Definition 1. While these are isometric to each other, we used the models in Definition 1 because we can formulate the dissimilarity function as a simple function of a linear combination of inner products $\boldsymbol{z}^\top \boldsymbol{z}$, $\boldsymbol{z}^\top \boldsymbol{z}'$, and $\boldsymbol{z}'^\top \boldsymbol{z}'$, or $\langle \boldsymbol{x}, \boldsymbol{x}' \rangle_{\mathrm{M}}$, which makes it easy to apply techniques in [30, 21].

**Graph Embedding**  Consider a graph $(\mathcal{V}, \mathcal{E})$, where $\mathcal{V}$ is the vertex set and $\mathcal{E} \in \mathcal{V} \times \mathcal{V}$ is the edge set. Here, we only consider undirected graphs, and thus assume that if $(i, j) \in \mathcal{E}$ then $(j, i) \in \mathcal{E}$ holds. Graph embedding for a graph $(\mathcal{V}, \mathcal{E})$ is a special case of embedding problem defined above, where the vertices $\mathcal{V}$ are the entities, the true dissimilarity between two entities $i, j$ is given by the graph distance defined by

$$\delta^*(i, j) := \min \left\{ K \mid \exists (v_1, v_2, \ldots, v_{K-1}) \in \mathcal{V}^{K-1}, (i, v_1), (v_1, v_2), \ldots, (v_{K-1}, j) \in \mathcal{E}. \right\}, \quad (4)$$

and $\theta^* = 1$. Then, the objective defined by (1) is equivalent to

$$(i, j) \in \mathcal{E} \Leftrightarrow \delta_{\mathcal{Z}}(z_i, z_j) \leq \theta_z. \qquad (5)$$

We say that a pair $(i, j)$ is *truly positive* if $(i, j) \in \mathcal{E}$ and *truly negative* otherwise.

In the following, we denote by $\deg(i)$ the degree of $i \in \mathcal{V}$, defined by $\deg(i) := \{j | (i, j) \in \mathcal{E}\}$.

## 2.2  Data distribution

In this paper, we consider negative-sampling-based training data and loss functions. Training data consist of $M$ positive-negative pair sequences. The $m$-th positive-negative pair sequence $s_m = (s_m^+, s_m^-)$ consists of $K^+$ positive pairs $s_m^+ := \left( (i_{1,m}^+, j_{1,m}^+), (i_{2,m}^+, j_{2,m}^+), \ldots, (i_{K^+,m}^+, j_{K^+,m}^+) \right) \in (\mathcal{V} \times \mathcal{V})^{K^+}$ and $K^-$ negative pairs $s_m^- := \left( (i_{1,m}^-, j_{1,m}^-), (i_{2,m}^-, j_{2,m}^-), \ldots, (i_{K^-,m}^-, j_{K^-,m}^-) \right) \in (\mathcal{V} \times \mathcal{V})^{K^-}$. Here, for $m = 1, 2, \ldots, M$, we expect that $i_{k,m}^+$ and $j_{k,m}^+$ are similar to each other, that is, $\delta^*\left(i_{k,m}^+, j_{k,m}^+\right) \leq \theta^*$ is valid for $k^+ = 1, 2, \ldots, K^+$, and conversely, $i_{k,m}^-$ and $j_{k,m}^-$ are dissimilar to each other, that is, $\delta^*\left(i_{k,m}^-, j_{k,m}^-\right) > \theta^*$ is valid for $k^- = 1, 2, \ldots, K^-$, although these rules do not always hold owing to noise or the sampling strategy aiming to save the computational cost. To derive a meaningful generalization error bound, an assumption on the distribution of the training data is needed. We consider the following weak assumption.

**Assumption 1.** $s_1, s_2, \ldots, s_M$ are independently and identically distributed.

**Remark 2.** Assumption 1 does NOT imply the independence and identity of the distribution of pairs in $s_m^+$ and $s_m^-$. For example, the distribution of $\left(i_{k',m}^-, j_{k',m}^-\right)$ may depend on $\left(i_{k,m}^-, j_{k,m}^-\right)$. This weakness assumption allows us to discuss practical negative sampling settings where negative pairs' distributions depend on positive pairs.

We give some training data distribution examples, which may be noisy. In the following, $\mathbb{P}\left[\left(i_{k,m}^+, j_{k,m}^+\right)\right]$ and $\mathbb{P}\left[\left(i_{k,m}^-, j_{k,m}^-\right)\right]$ denote the probability of edge $\left(i_{k,m}^+, j_{k,m}^+\right)$ and $\left(i_{k,m}^-, j_{k,m}^-\right)$ being generated as the $k$-th positive and negative pairs, respectively, and $\mathbb{P}\left[\left(i_{k,m}^-, j_{k,m}^-\right)\Big|\left(i_{k',m}^+, j_{k',m}^+\right)\right]$ denotes the probability of edge $\left(i_{k,m}^-, j_{k,m}^-\right)$ being generated as the $k$-th negative pair given $\left(i_{k',m}^+, j_{k',m}^+\right)$ being generated as the $k'$-th positive pair.

**Example 1.** (Data Distribution)

**(a)** (Simple positive-negative sampling) First, we consider the simplest case. Regarding the positive pair generation, suppose that all the pairs $(i, j)$ in the edge set $\mathcal{E}$ have the same probability of being generated as a positive pair, as a simple case. Also, suppose that all the pairs $(i, j)$ not in the edge set have a possibly non-zero probability that is lower than that for the positive pairs, which means the existence of noise. Here, we assume that each pair not in the edge set have the same probability, that is, the following holds:

$$\mathbb{P}\left[\left(i_{k,m}^+, j_{1,m}^+\right)\right] = \begin{cases} p^+ & \text{if } \left(i_{k,m}^+, j_{k,m}^+\right) \in \mathcal{E}, \\ r^+ p^+ & \text{if } \left(i_{k,m}^+, j_{k,m}^+\right) \notin \mathcal{E} \text{ and } i_{k,m}^+ \neq j_{k,m}^+, \\ 0 & \text{if } i_{k,m}^+ = j_{k,m}^+, \end{cases} \tag{6}$$

where $r^+ \in [0, 1]$ indicates the noise intensity. If $r = 0$, then only truly positive pairs appear, and if $r = 1$, then all edges appears in the same probability. Here, since $\sum_{\left(i_{1,m}^+, j_{1,m}^+\right) \in \mathcal{V} \times \mathcal{V}} \mathbb{P}\left[\left(i_{1,m}^+, j_{1,m}^+\right)\right] = 1$, we have that $p^+ = \frac{1}{(1-r^+)|\mathcal{E}|+r^+|\mathcal{V}|(|\mathcal{V}|-1)}$. For negative pair sampling, we consider the following simple distribution.

$$\mathbb{P}\left[\left(i_{k,m}^-, j_{1,m}^-\right)\right] = \begin{cases} p^- & \text{if } \left(i_{k,m}^-, j_{k,m}^-\right) \notin \mathcal{E} \text{ and } i_{k,m}^- \neq j_{k,m}^-, \\ r^- p^- & \text{if } \left(i_{k,m}^-, j_{k,m}^-\right) \in \mathcal{E}, \\ 0 & \text{if } i_{k,m}^- = j_{k,m}^-, \end{cases} \tag{7}$$

where $r^- \in [0, 1]$ indicates the noise intensity. Also, $p^-$ is given by $p^- = \frac{1}{(1-r^-)|\mathcal{V}|(|\mathcal{V}|-1)+r^-|\mathcal{E}|}$.

**(b)** (Skipgram [1] type negative sampling) In some applications, it is not easy to sample truly negative pairs effectively. In this case, more effective but inaccurate methods are often used. In the following, we explain a negative sampling strategy in [1], a representative on in such methods. Let $K^+ = 1$, and consider the positive pair sampling strategy again (6). Based on this simple setting, we consider the negative sampling strategy in [1]. Here, one vertex of a negative pair $i_{k,m}^-$ is always the same as that of the positive pair $i_{1,m}^+$, and the other entity of the negative pair is generated according to the distribution whose probability mass function is proportional to a value $\pi\left(U\left(j_{1,m}^-\right)\right)$, where $\pi : \mathbb{R}_{\geq 0} \to \mathbb{R}_{\geq 0}$ is a function and $U\left(j_{1,m}^-\right)$ is the frequency of the vertex $j_{1,m}^-$ appearing in a positive pair. In summary, the conditional distribution is given by

$$\mathbb{P}\left[\left(i_{k,m}^-, j_{k,m}^-\right)\Big|\left(i_{1,m}^+, j_{1,m}^+\right)\right] = \begin{cases} q\pi\left(U\left(j_{1,m}^-\right)\right) & \text{if } i_{k,m}^- = i_{1,m}^+, \\ 0 & \text{otherwise.} \end{cases} \tag{8}$$

Here, since $\sum_{j_{k,m}^- \in \mathcal{V}} \mathbb{P}\left[\left(i_{k,m}^-, j_{k,m}^-\right)\Big|\left(i_{1,m}^+, j_{1,m}^+\right)\right] = 1$, we have $q = \frac{1}{\sum_{j_{k,m}^- \in \mathcal{V}} \pi\left(U\left(j_{1,m}^-\right)\right)}$. If $\mathbb{P}\left[\left(i_{1,m}^+, j_{1,m}^+\right)\right]$ is given by (6), $U\left(j_{1,m}^-\right)$ is given by $U\left(j_{1,m}^-\right) = \frac{(1-r^+)\deg\left(j_{1,m}^-\right)+r^+(|\mathcal{V}|-1)}{(1-r^+)|\mathcal{E}|+r^+|\mathcal{V}|(|\mathcal{V}|-1)}$. In the following, we set $\pi(x) = x$ for simplicity. Then we have $\mathbb{P}\left[\left(i_{k,m}^-, j_{k,m}^-\right)\Big|\left(i_{1,m}^+, j_{1,m}^+\right)\right] = U\left(j_{1,m}^-\right)$. The above setting is an example of the negative pair's distribution depending on the positive pair. Note that the above setting does not guarantee that truly negative pairs appear as a negative pair more frequently than truly positive pairs.

We remark that this paper's discussion is not limited to the above examples, and Assumption 1 is the only assumption for our main theorem.

## 2.3 Loss function

To quantify the consistency of the true dissimilarities of entities defined by $\delta^*$ and those of representations defined by $\delta_{\mathcal{Z}}$, we consider loss function $l : \mathbb{R}^{K^+} \times \mathbb{R}^{K^-} \to \mathbb{R}_{\geq 0}$. The loss on a positive-negative pair sequence $s_m$ is given by $l(\boldsymbol{\delta}_m^+, \boldsymbol{\delta}_m^-)$, where

$$\boldsymbol{\delta}_m^\bullet := \begin{bmatrix} \delta_{i_{1,m}^\bullet, j_{1,m}^\bullet} & \delta_{i_{2,m}^\bullet, j_{2,m}^\bullet} & \cdots & \delta_{i_{K^\bullet,m}^\bullet, j_{K^\bullet,m}^\bullet} \end{bmatrix}^\top, \tag{9}$$

for $\bullet = -, +$ and $\delta_{i,j}$ is given by $\delta_{i,j} := \delta_{\mathcal{Z}}(z_i, z_j)$ for $i, j \in \mathcal{V}$. Here, we expect that $l$ is increasing with respect to each element of $\boldsymbol{\delta}_m^+$, the dissimilarity between a positive pair, and decreasing with respect to each element of $\boldsymbol{\delta}_m^-$, the dissimilarity between a negative pair.

**Assumption 2.** The loss function $l$ is Lipschitz continuous for each variable.

We denote the Lipschitz constant of $l$ with respect to $\delta_{i_k^+, j_k^+}$ and $\delta_{i_k^-, j_k^-}$ by $L_k^+$ and $L_k^-$, respectively. Note that the assumption regarding the loss function's Lipschitz continuity is common to derive a generalization bound using statistical learning theory (e.g., [33]). The following examples show that the above framework is general enough to include the existing applications' settings. In the following, $h : \mathbb{R} \to \mathbb{R}$ is an increasing function that converts the dissimilarity.

**Example 2.** (Loss functions)

**(a)** (Sigmoid-base loss) Define $l$ by

$$l(\boldsymbol{\delta}_m^+, \boldsymbol{\delta}_m^-) := \sum_{k^+=1}^{K^+} \ln \sigma\left(-\left[h\left(\delta_{i_{k^+,m}^+, j_{k^+,m}^+}\right) - h(\theta)\right]\right) + \sum_{k^-=1}^{K^-} \ln \sigma\left(h\left(\delta_{i_{k^-,m}^-, j_{k^-,m}^-}\right) - h(\theta)\right), \tag{10}$$

where $\sigma$ is a sigmoid shape function. For example, we can use the standard sigmoid function defined by $\sigma_{\mathrm{std}}(x) = \frac{1}{1+\exp(-x)}$, the hinge loss function $\sigma_{\mathrm{hinge}}(x) = \max\{0, x+1\}$ or the ramp loss function $\sigma_{\mathrm{ramp}}(x) = \min\{1, \sigma_{\mathrm{hinge}}(x)\}$. $h$ is $L_h$-Lipschitz. Then $L_{k^+}^+ = L_{k^-}^- = \frac{L_h}{4}$ if $\sigma = \sigma_{\mathrm{std}}$ and $L_{k^+}^+ = L_{k^-}^- = L_h$ if $\sigma = \sigma_{\mathrm{hinge}}$ or $\sigma = \sigma_{\mathrm{ramp}}$, for $k^+ = 1, 2, \ldots, K^+$ and $k^- = 1, 2, \ldots, K^-$. The above loss function (10) with the standard sigmoid function $\sigma$ corresponds to the negative-sampling-based loss function in [1] ($h(x) = x$ and $\theta = 0$) and the loss function in [20] for network embedding ($h(x) = \frac{x}{t}$ and $\theta = r$, where $r, t$ are fixed constants defined in the paper).

**(b)** (Softmax-like loss) Let $K^+ = 1$ and define $l$ by

$$l(\boldsymbol{\delta}_m^+, \boldsymbol{\delta}_m^-) := -h\left(\delta_{i_{1,m}^+, j_{1,m}^+}\right) + \ln \sum_{k^-=1}^{K^-} \exp\left(-h\left(\delta_{i_{k^-,m}^-, j_{k^-,m}^-}\right)\right). \tag{11}$$

Suppose that $h$ is $L_h$-Lipschitz. Then $L_{k^+}^+ = L_{k^-}^- = L_h$, for $k^+ = 1, 2, \ldots, K^+$ and $k^- = 1, 2, \ldots, K^-$. The loss function (11) corresponds to the loss function for embedding taxonomy in [20] with $h(x) = \sqrt{x}$, although this $h$ gives a non-Lipschitz-continuous loss function. In this paper, since a non-smooth loss function often loses a generalization guarantee, we only consider the case where $h$ and $l$ are Lipschitz continuous, as in [33].

## 2.4 Generalization error

The core discussion of this paper is the generalization error, which is the difference between the empirical risk and expected risk. We define the empirical risk function $\hat{\mathcal{R}}_{(s_m)_{m=1}^M}^{\mathcal{Z}} : (\mathcal{Z})^V \to \mathbb{R}_{\geq 0}$ on training data $s = (s_m)_{m=1}^M$ and the expected risk function $\mathcal{R}^{\mathcal{Z}} : (\mathcal{Z})^V \to \mathbb{R}_{\geq 0}$ as follows:

$$\hat{\mathcal{R}}_s^{\mathcal{Z}}\left((z_v)_{v=1}^V\right) := \frac{1}{M} \sum_{m=1}^M l(\boldsymbol{\delta}_m^+, \boldsymbol{\delta}_m^-), \quad \mathcal{R}^{\mathcal{Z}}\left((z_v)_{v=1}^V\right) := \mathbb{E}_{s_m} l(\boldsymbol{\delta}_m^+, \boldsymbol{\delta}_m^-). \tag{12}$$

Let $\mathcal{B} \subset \mathcal{Z}^{|\mathcal{V}|}$ be the space which we search for optimal representations $z_1, z_2, \ldots, z_{|\mathcal{V}|}$. As we discuss in the next section, $\mathcal{B}$ may be a bounded set rather than the whole space $\mathcal{Z}^{|\mathcal{V}|}$. We define the empirical risk minimizer $(\hat{z}_v)_{v=1}^{|\mathcal{V}|}$ and expected risk minimizer $(z_v^*)_{v=1}^{|\mathcal{V}|}$ by

$$(\hat{z}_v)_{v=1}^{|\mathcal{V}|} := \underset{(z_v)_{v=1}^{|\mathcal{V}|} \in \mathcal{B}}{\mathrm{argmin}} \ \hat{\mathcal{R}}_s^{\mathcal{Z}}\Big((z_v)_{v=1}^{|\mathcal{V}|}\Big), \quad (z_v^*)_{v=1}^{|\mathcal{V}|} := \underset{(z_v)_{v=1}^{|\mathcal{V}|} \in \mathcal{B}}{\mathrm{argmin}} \ \mathcal{R}^{\mathcal{Z}}\Big((z_v)_{v=1}^{|\mathcal{V}|}\Big). \tag{13}$$

Our interest is the *excess risk* given by $\mathcal{R}^{\mathcal{Z}}\Big((\hat{z}_v)_{v=1}^{|\mathcal{V}|}\Big) - \mathcal{R}^{\mathcal{Z}}\Big((z_v^*)_{v=1}^{|\mathcal{V}|}\Big)$, which indicates the generalization error of embedding. We derive the upper bound of the excess risk in the next section.

**Remark 3.** In the remainder of this paper, we compare the generalization error among multiple dissimilarity spaces. Since the loss itself depends on the embedding space's dissimilarity function $\mathcal{Z}$, the comparison is not always completely fair. Nevertheless, the comparison can be meaningful. For example, consider the 0-1 loss defined by (10) with $\sigma(x) = \sigma_{0-1}(x) := \begin{cases} 1 & \text{if } x \geq 0, \\ 0 & \text{if } x < 0. \end{cases}$ Regardless of the choice of embedding space $\mathcal{Z}$, the risk function on the 0-1 loss indicates the error rate of classifying a pair $(i,j)$ into "similar ($\delta^*(i,j) \leq \theta^*$)" or "dissimilar ($\delta^*(i,j) > \theta^*$)." Thus, it is fair to compare this 0-1-loss-based risk between different embedding spaces. Although the 0-1 loss itself does not satisfy Assumption 2, since the hinge loss and ramp loss dominates the 0-1 loss in that $\sigma_{0-1}(x) \leq \sigma_{\mathrm{ramp}}(x) \leq \sigma_{\mathrm{hinge}}(x)$, deriving and comparing bounds for these loss indirectly enables comparison between the 0-1-loss-based risks of different embedding spaces. In this sense, comparing the risks of different embedding spaces based on a Lipschitz continuous loss is of interest.

# 3 Generalization Bounds for Graph Embedding

## 3.1 Assumptions on Embedding Space's Radius

As also discussed in [21] for ordinal embedding, to derive a finite generalization bound, in general, it is necessary to restrict parameters (in embedding cases, representations) to a bounded domain (e.g., linear prediction models [37, 38], neural networks [37, 39]). In this section, we discuss our restriction on embedding space. For the derived generalization bound to be practical, the restriction should be simple and geometrically intuitive. Following [21], we put the following simple restrictions on embedding space's radius. Specifically, we discuss the case where we search for representations in $\mathcal{B} = \mathcal{B}_R$ defined by

$$\mathcal{B}_R := \Big\{ (z_v)_{v=1}^{|\mathcal{V}|} \Big| \forall v \in \mathcal{V} : \Delta_{\mathcal{Z}}(z_0, z_v) \leq R \Big\}, \tag{14}$$

where $z_0$ is $[0 \quad 0 \quad \ldots 0] \in \mathbb{R}^D$ for $\mathcal{Z} = \mathbb{R}^D, \mathbb{I}^D$ and $[1 \quad 0 \quad \ldots 0] \in \mathbb{R}^{D+1}$ for $\mathcal{Z} = \mathbb{L}^D, \mathbb{S}^D$, and $\Delta_{\mathbb{I}^D}$ is defined by $\Delta_{\mathbb{I}^D}(z, z') = \|z' - z\|_2$. In the next section, we provide the generalization error bound for empirical risk minimizer in $\mathcal{B}^R$.

## 3.2 Main Result: Finite Sample Upper Bounds for the Generalization Error

In this section, we first give our main theorem, which gives an upper bound for the generalization error, followed by remarks about intuitive interpretation of the bound and simplified versions.

**Theorem 1.** *Let $\mathcal{Z} = \mathbb{R}^D, \mathbb{L}^D, \mathbb{S}^D$ or $\mathbb{I}^D$, and $(\hat{z}_v)_{v=1}^{|\mathcal{V}|}$ and $(z_v^*)_{v=1}^{|\mathcal{V}|}$ be empirical and expected risk minimizers defined by* (13). *Under Assumptions 1 and 2, the following inequality holds with probability $1 - \mathfrak{d}$:*

$$\mathcal{R}^{\mathcal{Z}}\Big((\hat{z}_v)_{v=1}^{|\mathcal{V}|}\Big) - \mathcal{R}^{\mathcal{Z}}\Big((z_v^*)_{v=1}^{|\mathcal{V}|}\Big) \leq \frac{2\omega_{\mathcal{Z}}(R)}{M} \sum_{\bullet=+,-} \sum_{k=1}^{K^\bullet} L_k^\bullet \Big(\sqrt{2M\nu_{k\mathcal{Z}}^\bullet \ln|\mathcal{V}|} + \frac{\kappa_{\mathcal{Z}}}{3}\ln|\mathcal{V}|\Big) + I_l(\mathcal{B})\sqrt{\frac{\ln\frac{2}{\mathfrak{d}}}{M}}, \tag{15}$$

*where $\omega_{\mathcal{Z}}$ is defined by $\omega_{\mathbb{R}^D}(R) = \omega_{\mathbb{I}^D}(R) := (2R)^2$, $\omega_{\mathbb{L}^D}(R) := \cosh^2 R + \sinh^2 R$ and $\omega_{\mathbb{S}^D}(R) := \frac{2\arccos(1-\cos 2R)}{\sqrt{-\cos 2R(\cos 2R - 2)}}$, $\kappa_{\mathbb{L}^D} = \kappa_{\mathbb{S}^D} = \kappa_{\mathbb{I}^D} = \frac{1}{2}$ and $\kappa_{\mathbb{R}^D} = 2$, and $\nu_k^\bullet :=$*

$$\left\|\mathbb{E}_{\left(i^{\bullet}_{k,m}, j^{\bullet}_{k,m}\right)} \boldsymbol{E}^{\mathcal{Z}}_{i^{\bullet}_{k,m}, j^{\bullet}_{k,m}}{}^{\top} \boldsymbol{E}^{\mathcal{Z}}_{i^{\bullet}_{k,m}, j^{\bullet}_{k,m}}\right\|_{\mathrm{op},2}\right. \text{ for } k = 1, 2, \ldots, K^{\bullet} \text{ and } \bullet = +, -. \text{ Here, } \boldsymbol{E}^{\mathcal{Z}}_{i,j} \text{ is}$$
defined by

$$\left[\boldsymbol{E}^{\mathcal{Z}}_{i,j}\right]_{i',j'} := \begin{cases} a^{\mathcal{Z}}_{diag} & \text{if } (i,j) = (i',j'), \\ a^{\mathcal{Z}}_{off} & \text{if } i' = j' = i \text{ or } i' = j' = j \text{ and } (i,j) \neq (i',j''), \\ 0 & \text{otherwise}, \end{cases} \tag{16}$$

where $a^{\mathbb{R}^D}_{diag} = 1$, $a^{\mathbb{R}^D}_{off} = -1$, $a^{\mathbb{L}^D}_{diag} = -\frac{1}{2}$, $a^{\mathbb{S}^D}_{diag} = a^{\mathbb{I}^D}_{diag} = \frac{1}{2}$, and $a^{\mathbb{L}^D}_{off} = a^{\mathbb{S}^D}_{off} = a^{\mathbb{I}^D}_{off} = 0$. $I_l(\mathcal{B})$ is the range of $l$ defined by

$$I_l(\mathcal{B}) := \max\left\{ l(\boldsymbol{\delta}^+, \boldsymbol{\delta}^-) \mid s_m \in \mathcal{S}, (\boldsymbol{z}_v)_{v=1}^V \in \mathcal{B} \right\} - \min\left\{ l(\boldsymbol{\delta}^+, \boldsymbol{\delta}^-) \mid s_m \in \mathcal{S}, (\boldsymbol{z}_v)_{v=1}^V \in \mathcal{B} \right\}, \tag{17}$$

where $\mathcal{S} := (\mathcal{V} \times \mathcal{V})^{K^+} \times (\mathcal{V} \times \mathcal{V})^{K^-}$.

**Remark 4.** $\omega_{\mathcal{Z}}(R)$ corresponds to the embedding space's volume. Similar to ordinal embedding cases in [21], Theorem 1 argues that the larger the embedding space is, the larger the generalization error is. Since the volume of a ball is polynomial and exponential with respect to its radius in an inner product space and hyperbolic space, respectively, the generalization error also behaves correspondingly. $\nu^{\bullet}_k$ indicates the imbalance in the training data's distribution. Theorem 1 suggests that an imbalanced distribution causes large generalization error. This is intuitive because training data's imbalance makes it difficult to understand the big picture of the data. We will explore this in Example 3 and Remark 8.

**Remark 5.** If $\mathcal{Z} = \mathbb{R}^D$ or $\mathbb{I}^D$, multiplying a constant $\alpha$ to $R$ and $\alpha^2$ to Lipschitz constant $L^{\bullet}_k$ imply the same relaxation of the condition, since these two are equivalent by scaling representations. Hence, the bound in Theorem 1 for $\mathcal{Z} = \mathbb{R}^D$ or $\mathbb{I}^D$ includes a term $R^2 L^{\bullet}_k$, which indicates the essential size of the embedding space. This discussion is not true of $\mathcal{Z} = \mathbb{L}^D$ and $\mathbb{S}^D$, because balls with different sizes are not similar to each other in these spaces.

**Remark 6.** Owing to Assumption 2, $I_l \leq (2R)^2 \sum_{\bullet=+,-} \sum_{k=1}^{K^{\bullet}} L^{\bullet}_k$ always holds.

**Remark 7.** For any distribution on $s$, by Jensen's inequality, we have that $\nu^{\bullet}_k \leq \mathbb{E}_{\left(i^{\bullet}_{k,m}, j^{\bullet}_{k,m}\right)} \left\|\boldsymbol{E}^{\mathcal{Z}}_{i^{\bullet}_{k,m}, j^{\bullet}_{k,m}}{}^{\top} \boldsymbol{E}^{\mathcal{Z}}_{i^{\bullet}_{k,m}, j^{\bullet}_{k,m}}\right\|_{\mathrm{op},2} = c_{\mathcal{Z}}$ holds, where $c_{\mathcal{Z}} = 4$ for $\mathcal{Z} = \mathbb{R}^D$ and $c_{\mathcal{Z}} = \frac{1}{4}$ for $\mathcal{Z} = \mathbb{L}^D, \mathbb{S}^D$, and $\mathbb{I}^D$.

The proof is based on evaluating the Rademacher complexity [31, 32, 33], but technically nontrivial due to the dependency among the data. We have overcome this difficulty by our novel decomposition of the Rademacher complexity. See the supplementary materials for the complete proof of Theorem 1. In the following, we discuss examples of specific negative sampling strategies.

**Example 3** (Calculation of $\nu^{\bullet}_k$). Consider the setting in Example 1. Then the following holds.

**Proposition 1.**

*(a) Suppose that the data distribution is given by* (6) *and* (7). *Then we have that*

$$\nu^+_{k,\mathcal{Z}} = \frac{1}{2} \cdot \frac{(1-r^+)\max_{i\in\mathcal{V}} \deg(i) + r^+(|\mathcal{V}|-1)}{[(1-r^+)|\mathcal{E}| + r^+|\mathcal{V}|(|\mathcal{V}|-1)]} \leq \frac{1}{2} \cdot \frac{\max_{i\in\mathcal{V}} \deg(i)}{|\mathcal{E}|}, \tag{18}$$

$$\nu^-_{k,\mathcal{Z}} = \frac{1}{2} \cdot \frac{(1-r^-)[(|\mathcal{V}|-1) - \min_{i\in\mathcal{V}} \deg(i)] + r^-(|\mathcal{V}|-1)}{[(1-r^-)[|\mathcal{V}|(|\mathcal{V}|-1) - |\mathcal{E}|] + r^-|\mathcal{V}|(|\mathcal{V}|-1)]} \leq \frac{1}{2} \cdot \frac{(|\mathcal{V}|-1) - \min_{i\in\mathcal{V}} \deg(i)}{|\mathcal{V}|(|\mathcal{V}|-1) - |\mathcal{E}|}, \tag{19}$$

*for $\mathcal{Z} = \mathbb{L}^D, \mathbb{I}^D, \mathbb{S}^D$, and for $\mathcal{Z} = \mathbb{R}^D$ we have that*

$$\nu^+_{k,\mathbb{R}^D} \leq \frac{2[(1-r^+)\{2\max_{i\in\mathcal{V}} \deg(i)\} + r^+|\mathcal{V}|]}{(1-r^+)|\mathcal{E}| + r^+|\mathcal{V}|(|\mathcal{V}|-1)} \leq 4 \cdot \frac{\max_{i\in\mathcal{V}} \deg(i)}{|\mathcal{E}|}, \tag{20}$$

$$\nu^-_{k,\mathbb{R}^D} \leq 4\frac{1}{|\mathcal{V}|} \frac{(|\mathcal{V}|-1) - (1-r^-)\min_{i\in\mathcal{V}} \deg(i)}{(|\mathcal{V}|-1) - (1-r^-)\frac{|\mathcal{E}|}{|\mathcal{V}|}} \leq 4 \cdot \frac{1}{|\mathcal{V}|} \frac{(|\mathcal{V}|-1) - \min_{i\in\mathcal{V}} \deg(i)}{(|\mathcal{V}|-1) - \frac{|\mathcal{E}|}{|\mathcal{V}|}}. \tag{21}$$

*(b) If the conditional distribution of negative pairs is given by* (8), *then we have that*

$$\nu^-_{k,\mathcal{Z}} = \frac{1}{2} \cdot \frac{(1-r^-)\max_{i\in\mathcal{V}} \deg(i) + r^-(|\mathcal{V}|-1)}{[(1-r^-)|\mathcal{E}| + r^-|\mathcal{V}|(|\mathcal{V}|-1)]} \leq \frac{1}{2} \cdot \frac{\max_{i\in\mathcal{V}} \deg(i)}{|\mathcal{E}|}, \tag{22}$$

*for* $\mathcal{Z} = \mathbb{L}^D, \mathbb{I}^D, \mathbb{S}^D$, *and*

$$\nu^-_{k,\mathbb{R}^D} = 2\left[\frac{(1-r^-)\max_{i\in\mathcal{V}}\deg(i) + r^-(|\mathcal{V}|-1)}{(1-r^-)|\mathcal{E}| + r^-|\mathcal{V}|(|\mathcal{V}|-1)} + 1\right] \leq 2\left[\frac{\max_{i\in\mathcal{V}}\deg(i)}{|\mathcal{E}|} + 1\right]. \quad (23)$$

See the supplementary materials for the proof of Proposition 1.

**Remark 8.** We can interpret the evaluations of $\nu_1^+$ and $\nu_k^-$ in Example 3 as follows. In the following, we take (18) and (22) as examples, but similar discussion holds for all settings discussed in Example 3. First, we consider $r = 0$, which implies no noise setting. In this case, $\nu_1^+ = \nu_k^- = \frac{1}{2} \cdot \frac{\max_{i\in\mathcal{V}}\deg(i)}{|\mathcal{E}|}$ is valid. Here, the right hand side indicates the edge distribution's imbalance. For example, the right hand side takes the minimum $\frac{1}{2|\mathcal{V}|}$ if the graph is regular, where the distribution of edges are completely balanced, and takes the maximum $\frac{1}{2}$ if the graph is a star, where all edges are connected to one vertex, which is most imbalanced case. Hence, the equation regarding $\nu_1^+$ and $\nu_k^-$ implies that the more balanced the edge distribution is, the smaller generalization error it has. This relation is consistent to what we have emphasized in Remark 4. Second, we consider the relation between $r$ and the generalization error. The bound is decreasing with respect to $r$, the noise intensity. This is because training data is balanced between truly positive and negative pairs if the noise is intensive. Note that it does not imply that larger noise is better, because large noise worsens the optimal expected loss, even though it has small generalization error, which may lead to a large loss of learned representations.

## 4 Comparison between LGE and HGE

Theorem 1 implies that HGE has a larger generalization error than LGE; in other words, HGE has a higher variance than LGE. Conversely, hyperbolic space's ability to obtain low-distortion representations for a complete noiseless tree has been shown [29, 40, 21]; in other words, HGE has a lower bias than LGE. To evaluate the performance of learning models, discussing the bias-variance trade-off is essential. In this section, by combining the above bias and variance discussions, we derive an explicit condition on which HGE outperforms LGE in obtaining representations for a tree.

For fair discussion, suppose that the loss function $l$ is given by (10) with the ramp loss function $\sigma = \sigma_{\mathrm{ramp}}$, which dominates the 0-1 loss function as discussed in Remark 3, and the data distribution is given by the setting in Example 1 (a) with $K^+ = K^- = 1$. Consider the following conditions stronger than (5):

$$(i,j) \in \mathcal{E} \Rightarrow \delta_{\mathcal{Z}}(z_i, z_j) \leq \theta_{\mathcal{Z}} - 1, \quad (i,j) \notin \mathcal{E} \Rightarrow \delta_{\mathcal{Z}}(z_i, z_j) \geq \theta_{\mathcal{Z}} + 1, \quad (24)$$

and define $\mathcal{R}^*$ as the expected risk given by representations that satisfy the above conditions. Under the conditions, the loss is zero if truly positive pairs appear as positive pair $\left(\left(i^+_{k,m}, j^+_{k,m}\right) \in \mathcal{E}\right)$ or truly negative pairs appear as negative pair $\left(\left(i^-_{k,m}, j^-_{k,m}\right) \in \mathcal{E}\right)$. Then $\mathcal{R}^*$ is the achievable minimum expected risk.

Let $V_{\min}^{\mathbb{R}^2}$ and $V_{\min}^{\mathbb{L}^2}$ denote the minimum $V$ attainable by LGE using $\mathbb{R}^2$ and HGE using $\mathbb{L}^2$, respectively. Assume that the true dissimilarity $\delta^*$ is the graph distance of a tree. Regarding $V_{\min}^{\mathbb{R}^2}$ and $V_{\min}^{\mathbb{L}^2}$, the following lemmata hold [29, 40, 21] (See the supplementary materials for the proofs).

**Lemma 1.** *Suppose that $(\mathcal{V}, \mathcal{E})$ is a tree and $\delta^* : \mathcal{V} \times \mathcal{V} \to \mathbb{R}_{\geq 0}$ is given by its graph distance. Then, there exist $R \in \mathbb{R}_{\geq 0}$, representations $(z_1, z_2, \ldots, z_V) \in \mathcal{B}_R$ in $\mathbb{L}^2$, and threshold $\theta_{\mathcal{Z}} \in \mathbb{R}$ that satisfy (24) for all $i, j \in \mathcal{V}$. In particular, $\mathcal{R}^{\mathcal{Z}}\left((z_v^*)_{v=1}^{|\mathcal{V}|}\right) = \mathcal{R}^*$.*

**Lemma 2.** *Let $(\mathcal{V}, \mathcal{E})$ be a graph and define $W$ and $\rho$ by $W := |\mathcal{V}|(|\mathcal{V}|-1)$ and $\mathcal{E} := \frac{|\mathcal{E}|}{W}$. Define $\mu := \min\{\mu^+, \mu^-\}$, where $\mu^+ := \frac{1}{W}\frac{1}{(1-r^+)\rho+r^+} - \frac{1}{W}\frac{r^-}{(1-r^-)(1-\rho)+r^-}$ and $\mu^- := \frac{1}{W}\frac{1}{(1-r^-)(1-\rho)+r^-} - \frac{1}{W}\frac{r^+}{(1-r^+)\rho+r^+}$. In LGE, the expected risk of the expected risk minimizers satisfies $\mathcal{R}^{\mathcal{Z}}\left((z_v^*)_{v=1}^{|\mathcal{V}|}\right) \geq \mathcal{R}^* + V_{\min}^{\mathbb{R}^2}\mu$. Here, $V_{\min}^{\mathbb{R}^2}$ is not smaller than the number of disjoint 6-star subgraphs in the graph for the 2-dimensional LGE.*

By Theorem 1, we can conclude as follows:

**Proposition 2.** *Suppose that $(\mathcal{V}, \mathcal{E})$ is a tree and $\delta^* : \mathcal{V} \times \mathcal{V} \to \mathbb{R}_{\geq 0}$ is given by its graph distance, and take $R$ given in Lemma 1. Let $\nu_{1,\mathbb{L}^D} := \left( \sqrt{\nu_{1,\mathbb{L}^D}^+} + \sqrt{\nu_{1,\mathbb{L}^D}^-} \right)^2$. If*

$$M > \left( \frac{3\omega_{\mathbb{L}^D}(R)}{4|\mathcal{V}|\mu V_{\min}^{\mathbb{R}^2}} \left( \sqrt{8\nu_{1,\mathbb{L}^D} \ln |\mathcal{V}|} + \sqrt{\ln \frac{2}{\mathfrak{d}}} \right) + \frac{1}{2\left( \sqrt{8\nu_{1,\mathbb{L}^D} \ln |\mathcal{V}|} + \sqrt{\ln \frac{2}{\mathfrak{d}}} \right)} \right)^2, \quad (25)$$

*then HGE's $\mathcal{R}^{\mathcal{Z}}\left( (\hat{z}_v)_{v=1}^{|\mathcal{V}|} \right)$ is smaller than LGE's.*

See the supplementary materials for the proof of Proposition 2.

**Remark 9.** Proposition 2 implies that if the true dissimilarity is given by the graph distance of a tree, then HGE is better than LGE even if the data is not complete and noisy, if $M$ is larger than the right hand side of (25).

**Example 4.** We consider the complete balanced $\lambda$-ary tree with height $h$. Suppose $\lambda = 5$ and $h = 4$, and the positive and negative pair distributions are given by Example 1 (a) with $r^+ = r^- = 10^{-4}$. Then, HGE's $\mathcal{R}^{\mathcal{Z}}\left( (\hat{z}_v)_{v=1}^{|\mathcal{V}|} \right)$ with $R = 39.50...$ is smaller than LGE's if $M > 7.735 \times 10^{69}$ with probability 1 - $\mathfrak{d}$, where $\mathfrak{d} = 10^{-1}$.

See the supplementary materials for the proof of Example 4. Example 4 is the first specific calculation result that theoretically guarantees the superiority of HGE to LGE. Nevertheless, tightening the right hand side or deriving a necessary condition could be future work.

## 5 Conclusion

We have shown that LGE and HGE cause a polynomial and exponential error with respect to the embedding space's radius, respectively, and that imbalanced data distribution can worsen the error. Our bias-variance trade-off discussion implies that even though HGE has larger generalization error than LGE, HGE with sufficiently large number of data can represent hierarchical data more effectively. This discussion provides a guide for embedding space selection in real applications.

One limitation of our result is that it does not clarify the error's dependency on the dimension $D$. Our bound does not depend on $D$, which is not consistent to our intuition that using low-dimensional space should give low generalization error. This problem is essentially the same as that pointed out in [21], which partially used similar techniques to ours. Deriving a tighter bound in terms of the dimension for general ordinal embedding could be future work.

## Acknowledgments and Disclosure of Funding

This work was partially supported by JST KAKENHI 191400000190, 19K20337, and JST-AIP JPMJCR19U4, Daiwa Foundation Award (Ref: 13849/14682), and JST-PRESTO.

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
