# Supplementary Materials for Generalization Error Bounds for Graph Embedding Using Negative Sampling: Linear vs Hyperbolic

## A   Notation

Let $D, V \in \mathbb{Z}_{>0}$. We denote by $\mathrm{Sym}^V$ the set of $V \times V$ symmetric matrices, respectively. For a vector $\boldsymbol{x} \in \mathbb{R}^D$, we denote by $\|\boldsymbol{x}\|_2, \|\boldsymbol{x}\|_1$ the 2-norm and 1-norm of $\boldsymbol{x}$, defined by $\|\boldsymbol{x}\|_2 := \sqrt{\boldsymbol{x}^\top \boldsymbol{x}}$ and $\|\boldsymbol{x}\|_1 := \sum_i |x|_i$, respectively. For matrices $\boldsymbol{A}, \boldsymbol{B} \in \mathbb{R}^{D,V}$, we denote by $\langle \boldsymbol{A}, \boldsymbol{B} \rangle_{\mathrm{F}}$ the Frobenius inner-product of $\boldsymbol{A}$ and $\boldsymbol{B}$, defined by $\mathrm{Tr}\left(\boldsymbol{A}^\top \boldsymbol{B}\right)$, and denote by $\|\boldsymbol{A}\|_{2,\mathrm{op}}$ the operator norm with respect to the 2-norm defined by $\|\boldsymbol{A}\|_2 := \sum_{\boldsymbol{x} \in \mathbb{R}^V, \boldsymbol{x} \neq \boldsymbol{0}} \frac{\|\boldsymbol{A}\boldsymbol{x}\|_2}{\|\boldsymbol{x}\|_2}$. We denote by $\|\boldsymbol{A}\|_*$ the nuclear norm of $\boldsymbol{A}$ defined by the sum of singular values. For symmetric matrices $\boldsymbol{A}, \boldsymbol{B} \in \mathrm{Sym}^V$, we write $\boldsymbol{A} \succeq \boldsymbol{B}$ if $\boldsymbol{A} - \boldsymbol{B}$ is positive semi-definite. For an integer $K \in \mathbb{Z}_{>0}$, a set $\mathcal{X}$, a function $f : \mathcal{X} \to \mathbb{R}^K$ on $\mathcal{X}$, and a function $l : \mathbb{R}^K \to \mathbb{R}$, we define the composition $l \circ f$ by $(l \circ f)(x) := l(f(x))$. Also, for a function set $\mathcal{F} \subset \{f \mid f : \mathcal{X} \to \mathbb{R}^K\}$, we define $l \circ \mathcal{F}$ by $\{l \circ f \mid f \in \mathcal{F}\}$.

## B   Proof of Theorem 1

In this section, we prove Theorem 1. We first recall the theorem.

**Theorem** (Theorem 1 in the body text). *Let $\mathcal{Z} = \mathbb{R}^D, \mathbb{L}^D, \mathbb{S}^D$ or $\mathbb{I}^D$, and $(\hat{\boldsymbol{z}}_v)_{v=1}^{|\mathcal{V}|}$ and $(\boldsymbol{z}_v^*)_{v=1}^{|\mathcal{V}|}$ be empirical and expected risk minimizers defined by* (13). *Under Assumptions 1 and 2, the following inequality holds with probability $1 - \mathfrak{d}$:*

$$\mathcal{R}^{\mathcal{Z}}\left((\hat{\boldsymbol{z}}_v)_{v=1}^{|\mathcal{V}|}\right) - \mathcal{R}^{\mathcal{Z}}\left((\boldsymbol{z}_v^*)_{v=1}^{|\mathcal{V}|}\right) \leq \frac{2\omega_{\mathcal{Z}}(R)}{M} \sum_{\bullet=+,-} \sum_{k=1}^{K^\bullet} L_k^\bullet\left(\sqrt{2M\nu_{k\,\mathcal{Z}}^\bullet \ln|\mathcal{V}|} + \frac{\kappa_{\mathcal{Z}}}{3}\ln|\mathcal{V}|\right) + I_l(\mathcal{B})\sqrt{\frac{\ln\frac{2}{\mathfrak{d}}}{M}},$$
(26)

*where $\omega_{\mathcal{Z}}$ is defined by $\omega_{\mathbb{R}^D}(R) = \omega_{\mathbb{I}^D}(R) := (2R)^2$, $\omega_{\mathbb{L}^D}(R) := \cosh^2 R + \sinh^2 R$ and $\omega_{\mathbb{S}^D}(R) := \frac{2\arccos(1 - \cos 2R)}{\sqrt{-\cos 2R(\cos 2R - 2)}}$, $\kappa_{\mathbb{L}^D} = \kappa_{\mathbb{S}^D} = \kappa_{\mathbb{I}^D} = \frac{1}{2}$ and $\kappa_{\mathbb{R}^D} = 2$, and $\nu_k^\bullet :=$ $\left\| \mathbb{E}_{(i_{k,m}^\bullet, j_{k,m}^\bullet)} \boldsymbol{E}_{i_{k,m}^\bullet, j_{k,m}^\bullet}^{\mathcal{Z}} {}^\top \boldsymbol{E}_{i_{k,m}^\bullet, j_{k,m}^\bullet}^{\mathcal{Z}} \right\|_{\mathrm{op},2}$ for $k = 1, 2, \ldots, K^\bullet$ and $\bullet = +, -$. Here, $\boldsymbol{E}_{i,j}^{\mathcal{Z}}$ is defined by*

$$\left[\boldsymbol{E}_{i,j}^{\mathcal{Z}}\right]_{i',j'} := \begin{cases} a_{diag}^{\mathcal{Z}} & \text{if } (i,j) = (i',j'), \\ a_{off}^{\mathcal{Z}} & \text{if } i' = j' = i \text{ or } i' = j' = j \text{ and } (i,j) \neq (i',j''), \\ 0 & \text{otherwise}, \end{cases}$$
(27)

*where $a_{diag}^{\mathbb{R}^D} = 1$, $a_{off}^{\mathbb{R}^D} = -1$, $a_{diag}^{\mathbb{L}^D} = -\frac{1}{2}$, $a_{diag}^{\mathbb{S}^D} = a_{diag}^{\mathbb{I}^D} = \frac{1}{2}$, and $a_{off}^{\mathbb{L}^D} = a_{off}^{\mathbb{S}^D} = a_{off}^{\mathbb{I}^D} = 0$. $I_l(\mathcal{B})$ is the range of $l$ defined by*

$$I_l(\mathcal{B}) := \max\left\{ l\left(\boldsymbol{\delta}^+, \boldsymbol{\delta}^-\right) \mid s_m \in \mathcal{S}, (\boldsymbol{z}_v)_{v=1}^V \in \mathcal{B} \right\} - \min\left\{ l\left(\boldsymbol{\delta}^+, \boldsymbol{\delta}^-\right) \mid s_m \in \mathcal{S}, (\boldsymbol{z}_v)_{v=1}^V \in \mathcal{B} \right\},$$
(28)

*where $\mathcal{S} := (\mathcal{V} \times \mathcal{V})^{K^+} \times (\mathcal{V} \times \mathcal{V})^{K^-}$.*

Our generalization error bound derivation is based on evaluation of the Rademacher complexity [31, 32, 33] of the loss function $l$ as a function of positive-negative pair sequence with representations as parameters. First, we define the Rademacher complexity below. Consider $\mathcal{F} \subset \{f | f : \mathcal{X} \to \mathbb{R}\}$, where $\mathcal{X}$ is the domain of the functions in the set $\mathcal{F}$. Let $M \in \mathbb{Z}_{>0}$ be the number of data points, and suppose that data points $x_1, x_2, \ldots, x_M \in \mathcal{X}$ are independently distributed according to some unknown fixed distribution $\mathcal{D}$. The Rademacher complexity of $\mathcal{F}$ is defined as follows:

**Definition 2.** Let $\sigma_1, \sigma_2, \ldots, \sigma_M$ be random values such that $\sigma_1, \sigma_2, \ldots, \sigma_M, x_1, x_2, \ldots, x_M$ are mutually independent and each of $\sigma_1, \sigma_2, \ldots, \sigma_M$ takes values $\{-1, +1\}$ with equal probability. The

Rademacher complexity $\mathfrak{R}_{\mathcal{D},M}(\mathcal{F})$ with respect to $\mathcal{D}$ is defined by

$$\mathfrak{R}_{\mathcal{D},M}(\mathcal{F}) := \mathbb{E}_{(x_m)_{m=1}^M} \mathbb{E}_{(\sigma_m)_{m=1}^M} \left[ \frac{1}{M} \sup_{f \in \mathcal{F}} \sum_{m=1}^M \sigma_m f(x_m) \right]. \tag{29}$$

By evaluating the Rademacher complexity, we can derive a generalization error bound by the following theorem provided by Bartlett and Mendelson [37] and arranged by Kakade *et al.* [38].

**Theorem 2** ([37, 38]). *Let $l : \mathcal{X} \to \mathbb{R}$ be a loss function. Define the empirical risk function $\hat{\mathcal{R}}_{(x_m)_{m=1}^M}(f)$ and expected risk function $\mathcal{R}(f)$ by*

$$\hat{\mathcal{R}}_{(x_m)_{m=1}^M}(f) := \frac{1}{M} \sum_{m=1}^M l((f(x_m))), \qquad \mathcal{R}(f) := \mathbb{E}_x \, l(f(x))). \tag{30}$$

*Assume that $l$ is $L_l$-Lipschitz continuous and bounded. Define*

$$I_l(\mathcal{F}) := \sup_{\substack{x \in \mathcal{X}, \\ f \in \mathcal{F}}} l(f(x)) - \inf_{\substack{x \in \mathcal{X}, \\ f \in \mathcal{F}}} l(f(x)). \tag{31}$$

*Then for any $\delta \in \mathbb{R}_{>0}$ and with probability at least $1 - \delta$ simultaneously for all $f \in \mathcal{F}$ we have that*

$$\mathcal{R}(f) - \hat{\mathcal{R}}_{(x_m)_{m=1}^M}(f) \leq 2\mathfrak{R}_{\mathcal{D},M}(\mathcal{F}) + I_l(\mathcal{F})\sqrt{\frac{\ln(1/\delta)}{2M}}. \tag{32}$$

From this theorem, we can easily derive an upper bound for the excess risk of the empirical risk minimizer as follows.

**Corollary 1.** *Define the empirical risk minimizer $\hat{f} \in \mathcal{F}$ and expected loss minimizer by $f^* \in \mathcal{F}$ by*

$$\hat{f} := \operatorname*{argmin}_{f \in \mathcal{F}} \hat{\mathcal{R}}_{(x_m)_{m=1}^M}(f), \quad f^* := \operatorname*{argmin}_{f \in \mathcal{F}} \mathcal{R}(f), \tag{33}$$

*and we call $\mathcal{R}\left(\hat{f}\right) - \mathcal{R}(f^*)$ the excess risk of $\hat{f}$. Then for any $\delta \in \mathbb{R}_{>0}$ and with probability at least $1 - \delta$ we have that*

$$\mathcal{R}\left(\hat{f}\right) - \mathcal{R}(f^*) \leq 2\mathfrak{R}_M(\mathcal{F}) + 2I_l(\mathcal{F})\sqrt{\frac{\ln(2/\delta)}{2M}}. \tag{34}$$

*Proof.* We have that

$$\begin{aligned}
&\mathcal{R}\left(\hat{f}\right) - \mathcal{R}(f^*) \\
&= \left(\mathcal{R}\left(\hat{f}\right) - \hat{\mathcal{R}}_{\mathcal{S}}\left(\hat{f}\right)\right) + \left(\hat{\mathcal{R}}_{\mathcal{S}}\left(\hat{f}\right) - \hat{\mathcal{R}}_{\mathcal{S}}(f^*)\right) + \left(\hat{\mathcal{R}}_{\mathcal{S}}(f^*) - \mathcal{R}(f^*)\right) \\
&\leq \left(\mathcal{R}\left(\hat{f}\right) - \hat{\mathcal{R}}_{\mathcal{S}}\left(\hat{f}\right)\right) + \left(\hat{\mathcal{R}}_{\mathcal{S}}(f^*) - \mathcal{R}(f^*)\right),
\end{aligned} \tag{35}$$

where the last inequality holds from the definition of $\hat{f}$. We complete the proof by evaluating the first and second term by Theorem 2 and Hoeffding's inequality, respectively. $\square$

Hence, it suffices to evaluate the Rademacher complexity of $l \circ \mathcal{F}_{\mathcal{B}_R^{\mathcal{Z}}}$, where $\mathcal{F}_{\mathcal{B}_R^{\mathcal{Z}}} \subset \left\{ f \mid f : (\mathcal{Z} \times \mathcal{Z})^{K^+} \times (\mathcal{Z} \times \mathcal{Z})^{K^-} \to \mathbb{R}^{K^+} \times \mathbb{R}^{K^-} \right\}$ is defined by

$$\mathcal{F}_{\mathcal{B}_R^{\mathcal{Z}}} := \left\{ f_{(z_v)_{v=1}^{|\mathcal{V}|}} \mid (z_v)_{v=1}^{|\mathcal{V}|} \in \mathcal{B}_R^{\mathcal{Z}} \right\}, \tag{36}$$

with $f_{(z_v)_{v=1}^{|\mathcal{V}|}} : (\mathcal{Z} \times \mathcal{Z})^{K^+} \times (\mathcal{Z} \times \mathcal{Z})^{K^-} \to \mathbb{R}^{K^+} \times \mathbb{R}^{K^-}$ defined by

$$f_{(z_v)_{v=1}^{|\mathcal{V}|}} \left( \left[ \begin{matrix} (i_1^+, j_1^+) \\ (i_2^+, j_2^+) \\ \vdots \\ (i_{K^+}^+, j_{K^+}^+) \end{matrix} \right], \left[ \begin{matrix} (i_1^-, j_1^-) \\ (i_2^-, j_2^-) \\ \vdots \\ (i_{K^-}^-, j_{K^-}^-) \end{matrix} \right] \right) := \left( \left[ \begin{matrix} \delta_{\mathcal{Z}}\left(z_{i_1^+}, z_{j_1^+}\right) \\ \delta_{\mathcal{Z}}\left(z_{i_2^+}, z_{j_2^+}\right) \\ \vdots \\ \delta_{\mathcal{Z}}\left(z_{i_{K^+}^+}, z_{j_{K^+}^+}\right) \end{matrix} \right], \left[ \begin{matrix} \delta_{\mathcal{Z}}\left(z_{i_1^-}, z_{j_1^-}\right) \\ \delta_{\mathcal{Z}}\left(z_{i_2^-}, z_{j_2^-}\right) \\ \vdots \\ \delta_{\mathcal{Z}}\left(z_{i_{K^-}^-}, z_{j_{K^-}^-}\right) \end{matrix} \right] \right). \tag{37}$$

However, the difficulty is that the loss function defined in Section 2.3 can be so complex that we cannot evaluate its Rademacher complexity directly in that it is the composition of multiple functions and the data distribution is not independent with respect to each element. Our following lemma overcomes this difficulty.

**Lemma 3.** *Suppose that* $l : \mathbb{R}^K \to \mathbb{R}$ *is a function Lipschitz continuous with respect to each element of the parameter with Lipschitz constant* $L_k$ *for* $k = 1, 2, \ldots, K$. *Let* $\mathcal{X}_1, \mathcal{X}_2, \ldots, \mathcal{X}_K$ *be sets. We consider a set* $\mathcal{F} \subset \{(f_1, f_2, \ldots, f_K) \mid f_k : \mathcal{X}_k \to \mathbb{R}, \text{ for } k = 1, 2, \ldots, K\}$ *of sequences of functions. Define a set of functions* $\mathcal{F}_k \subset \{f \mid f : \mathcal{X}_k \to \mathbb{R}\}$ *by*

$$\mathcal{F}_k := \left\{ f_k \middle| \begin{array}{l} \exists f_1, f_2, \ldots, f_{k-1}, f_{k+1}, f_{k+2}, \ldots, f_K, \\ (f_1, f_2, \ldots, f_{k-1}, f_k, f_{k+1}, f_{k+2}, \ldots, f_K) \in \mathcal{F} \end{array} \right\} \tag{38}$$

*Define a set* $l \circ \mathcal{F} \in \{g \mid g : \mathcal{X}_1, \mathcal{X}_2, \ldots, \mathcal{X}_K \to \mathbb{R}\}$ *of functions by*

$$l \circ \mathcal{F} := \left\{ \begin{array}{l} l(f_1, f_2, \ldots, f_K) : \mathcal{X}_1 \times \mathcal{X}_2 \times \cdots \times \mathcal{X}_K \to \mathbb{R}, \\ (x_1, x_2, \ldots, x_K) \mapsto l(f_1(x_1), f_2(x_2), \ldots, f_K(x_K)) \end{array} \middle| (f_1, f_2, \ldots, f_K) \in \mathcal{F} \right\}, \tag{39}$$

*Let* $\mathcal{D}$ *be a distribution on* $\mathcal{X}_1 \times \mathcal{X}_2 \times \ldots \mathcal{X}_K$, *and denote its marginal distribution on* $\mathcal{X}_k$ *by* $\mathcal{D}_k$ *for* $k = 1, 2, \ldots, K$. *Then the Rademacher complexity of* $l \circ \mathcal{F}$ *with respect to* $\mathcal{D}$ *is given by*

$$\mathfrak{R}_{\mathcal{D},M}(l \circ \mathcal{F}) = \sum_{k=1}^{K} L_k \mathfrak{R}_{\mathcal{D}_k,M}(\mathcal{F}_k). \tag{40}$$

We prove Lemma 3 later.

**Remark 10.** If $K = 1$, Lemma 3 is equivalent to the Ledoux-Talagrand contraction lemma [36].

In the following, we prove Theorem 1 using Lemma 3.

*Proof of Theorem 1.* Using Lemma 3, we can decompose the Rademacher complexity of $l \circ \mathcal{F}_{\mathcal{B}_R^{\mathcal{Z}}}$ as follows:

$$\mathfrak{R}_{\mathcal{D},M}\left(l \circ \mathcal{F}_{\mathcal{B}_R^{\mathcal{Z}}}\right) = \sum_{\bullet=-1,+1} \sum_{k=1}^{K^\bullet} L_k \mathfrak{R}_{\mathcal{D}_k^\bullet,M}\left(\mathcal{F}_{0,\mathcal{B}_R^{\mathcal{Z}}}\right), \tag{41}$$

where $\mathcal{F}_{0,\mathcal{B}_R^{\mathcal{Z}}} \subset \{f \mid f : \mathcal{Z} \times \mathcal{Z} \to \mathbb{R}\}$ is defined by

$$\mathcal{F}_{0,\mathcal{B}_R^{\mathcal{Z}}} := \left\{ f_{0,(z_v)_{v=1}^{|\mathcal{V}|}} \middle| (z_v)_{v=1}^{|\mathcal{V}|} \in \mathcal{B}_R^{\mathcal{Z}} \right\}, \tag{42}$$

with $f_{0,(z_v)_{v=1}^{|\mathcal{V}|}} : \mathcal{Z} \times \mathcal{Z} \to \mathbb{R}$ defined by

$$f_{0,(z_v)_{v=1}^{|\mathcal{V}|}}(i,j) := \delta_{\mathcal{Z}}(z_i, z_j). \tag{43}$$

Here, $\mathcal{D}$ is the distribution of the positive-negative pair sequence $\left(\left(i_{k,m}^+, j_{k,m}^+\right)_{k=1}^{K^+}, \left(i_{k,m}^-, j_{k,m}^-\right)_{k=1}^{K^-}\right)$, and $\mathcal{D}_k^\bullet$ is its marginal distribution with respect to $\left(i_{k,m}^\bullet, j_{k,m}^\bullet\right)$ where $\bullet = -, +$.

Therefore, it suffices to evaluate the Rademacher complexity of $\mathcal{F}_{0,\mathcal{B}_R^{\mathcal{Z}}}$. In the following, we achieve the evaluation by converting the function to that of the Gram matrix.

First, we introduce the Gram matrix. Let $\boldsymbol{z}_v$ be the coordinate vector of $z_v$. We define the Gram matrix $\boldsymbol{G}_{(z_v)_{v=1}^{|\mathcal{V}|}} \in \operatorname{Sym}^{|v|}$ of representations $(z_v)_{v=1}^{|\mathcal{V}|}$ by

$$\boldsymbol{G}_{(z_v)_{v=1}^{|\mathcal{V}|}} := \begin{bmatrix} \boldsymbol{z}_1 & \boldsymbol{z}_2 & \ldots & \boldsymbol{z}_{|\mathcal{V}|} \end{bmatrix}^\top \begin{bmatrix} \boldsymbol{z}_1 & \boldsymbol{z}_2 & \ldots & \boldsymbol{z}_{|\mathcal{V}|} \end{bmatrix} \tag{44}$$

if $(z_v)_{v=1}^{|\mathcal{V}|} \in \left(\mathbb{R}^D\right)^{|\mathcal{V}|}, \left(\mathbb{S}^D\right)^{|\mathcal{V}|}$, or $\left(\mathbb{I}^D\right)^{|\mathcal{V}|}$, and

$$\boldsymbol{G}_{(z_v)_{v=1}^{|\mathcal{V}|}} := \begin{bmatrix} \boldsymbol{z}_1 & \boldsymbol{z}_2 & \ldots & \boldsymbol{z}_{|\mathcal{V}|} \end{bmatrix}^\top \begin{bmatrix} -1 & & & & \\ & +1 & & & \\ & & +1 & & \\ & & & \ddots & \\ & & & & +1 \end{bmatrix} \begin{bmatrix} \boldsymbol{z}_1 & \boldsymbol{z}_2 & \ldots & \boldsymbol{z}_{|\mathcal{V}|} \end{bmatrix}, \tag{45}$$

if $(z_v)_{v=1}^{|\mathcal{V}|} \in (\mathbb{L}^D)^{|\mathcal{V}|}$.

For $\mathcal{S} \subset \mathrm{Sym}^{|\mathcal{V}|}$, we define

$$\mathcal{F}_{\mathcal{S}} := \{f_{\boldsymbol{S}} \mid \boldsymbol{S} \in \mathcal{S}\}, \tag{46}$$

where for $\boldsymbol{S} \in \mathrm{Sym}^{|\mathcal{V}|}$, we define $f_{\boldsymbol{S}} : \mathcal{V} \times \mathcal{V} \to \mathbb{R}$

$$f_{\boldsymbol{S}}(i, j) = \langle \boldsymbol{S}, \boldsymbol{E}_{i,j} \rangle_{\mathrm{F}}. \tag{47}$$

By the definition of $\delta_{\mathcal{Z}}$, we have that

$$f_{(z_v)_{v=1}^{|\mathcal{V}|}}(i, j) := \left( h_{\mathcal{Z}} \circ f_{\boldsymbol{G}_{(z_v)_{v=1}^{|\mathcal{V}|}}} \right)(i, j), \tag{48}$$

where

$$h_{\mathbb{R}^D}(x) = h_{\mathbb{I}^D}(x) := x, \qquad h_{\mathbb{S}^D}(x) := \arccos^2(x), \qquad h_{\mathbb{L}^D}(x) := \mathrm{arcosh}^2(x). \tag{49}$$

Define $\mathcal{G}_R \subset \mathrm{Sym}^{|\mathcal{V}|}$ by

$$\mathcal{G}_R^{\mathcal{Z}} := \left\{ \boldsymbol{G}_{(z_v)_{v=1}^{|\mathcal{V}|}} \,\middle|\, (z_v)_{v=1}^{|\mathcal{V}|} \in \mathcal{B}_R^{\mathcal{Z}} \right\}. \tag{50}$$

By applying the Ledoux-Talagrand contraction lemma [36] (or Lemma 3 with $K = 1$), we have that

$$\mathfrak{R}_{\mathcal{D}_k^\bullet, M}\left( \mathcal{F}_{0, \mathcal{B}_R^{\mathcal{Z}}} \right) \leq L_{h_{\mathcal{Z}}} \mathfrak{R}_{\mathcal{D}_k^\bullet, M}\left( \mathcal{F}_{\mathcal{G}_R^{\mathcal{Z}}} \right), \tag{51}$$

where $L_{h_{\mathcal{Z}}}$ is the Lipschitz constant of $h_{\mathcal{Z}}$. Here, we have that

$$L_{h_{\mathbb{R}^D}} = L_{h_{\mathbb{L}^D}} = L_{h_{\mathbb{I}^D}} = 1, \qquad L_{h_{\mathbb{S}^D}} = \frac{2 \arccos(1 - \cos 2R)}{\sqrt{-\cos 2R(\cos 2R - 2)}}. \tag{52}$$

Define $\mathcal{Q}_\lambda \subset \mathcal{S}^{|\mathcal{V}|}$ by

$$\mathcal{Q}_\lambda := \left\{ \boldsymbol{G} \in \mathrm{Sym}^{|\mathcal{V}|} \,\middle|\, \boldsymbol{G} \succeq \boldsymbol{O}, \|\boldsymbol{G}\|_* \leq \lambda. \right\}. \tag{53}$$

The following is valid.

$$\mathcal{G}_R^{\mathcal{Z}} \subset \begin{cases} \mathcal{Q}_{|\mathcal{V}|R^2} & \text{if } \mathcal{Z} = \mathbb{R}^D \text{ or } \mathbb{I}^D, \\ \mathcal{Q}_{|\mathcal{V}|} & \text{if } \mathcal{Z} = \mathbb{S}^D, \\ \mathcal{Q}_{|\mathcal{V}|(\cosh^2 R + \sinh^2 R)} & \text{if } \mathcal{Z} = \mathbb{L}^D. \end{cases} \tag{54}$$

The above result for $\mathcal{Z} = \mathbb{L}^D$ is the consequence of Lemma 8 in [21]. Those for $\mathcal{Z} = \mathbb{R}^D, \mathbb{S}^D$, and $\mathbb{I}^D$ are according to a basic property of the Gram matrix (e.g., Lemma 4 in [21]).

Therefore, evaluating $\mathfrak{R}_{\mathcal{D}, M}^{\mathrm{G}}(\mathcal{F}_{\mathcal{Q}_\lambda})$ completes the proof of Theorem 1. The following lemma gives $\mathfrak{R}_{\mathcal{D}, M}^{\mathrm{G}}(\mathcal{F}_{\mathcal{Q}_\lambda})$.

**Lemma 4.**

$$\mathfrak{R}_{\mathcal{D}_k^\bullet, M}(\mathcal{F}_{\mathcal{Q}_\lambda}) \leq \frac{\lambda}{M} \left[ \sqrt{2M\nu_k^\bullet \ln|\mathcal{V}|} + \frac{\kappa_{\mathcal{Z}}}{3} \ln|\mathcal{V}| \right], \tag{55}$$

*where*

$$\kappa_{\mathcal{Z}} := \begin{cases} +2 & \mathcal{Z} = \mathbb{R}^D, \\ +\frac{1}{2} & \mathcal{Z} = \mathbb{L}^D, \mathbb{S}^D, \mathbb{I}^D. \end{cases} \tag{56}$$

We obtain Theorem 1 by applying Lemma 4 to (54) that appears in (51). $\qquad \square$

The followings are proofs of the lemmata that appeared in the above proof.

*Proof of Lemma 4.* For $\lambda \in \mathbb{R}_{\geq 0}$, define $\mathcal{Q}_1^\lambda$ by

$$\mathcal{Q}_1^\lambda := \left\{ \lambda \boldsymbol{u} \boldsymbol{u}^\top \middle| \boldsymbol{u} \in \mathbb{R}^{|\mathcal{V}|}, \|\boldsymbol{u}\|_2 = 1. \right\}. \tag{57}$$

Since $\mathcal{Q}_1^\lambda$ is the convex hull of $\mathcal{Q}_\lambda$ and the Rademacher complexity of a function class equals that of its convex hull [37, Theorem 12-2], we have that

$$\mathfrak{R}_{\mathcal{D},M}^{\mathrm{G}}(\mathcal{F}_{\mathrm{G}}(\mathcal{Q}_\lambda)) = \mathfrak{R}_{\mathcal{D},M}^{\mathrm{G}}\big(\mathcal{F}_{\mathrm{G}}(\mathcal{Q}_1^\lambda)\big)$$

$$= \mathbb{E}_{((i_{k,m}^\bullet, j_{k,m}^\bullet))_{m=1}^M \sim (\mathcal{D}_k^\bullet)^M} \mathbb{E}_{\boldsymbol{\sigma}} \left[ \sup_{\|\boldsymbol{u}\|_2 \leq 1} \frac{1}{M} \sum_{m=1}^M \sigma_m \left\langle \lambda \boldsymbol{u} \boldsymbol{u}^\top, \boldsymbol{E}_{i_{k,m}^\bullet, j_{k,m}^\bullet}^{\mathcal{Z}} \right\rangle_{\mathrm{F}} \right]$$

$$= \mathbb{E}_{((i_{k,m}^\bullet, j_{k,m}^\bullet))_{m=1}^M \sim (\mathcal{D}_k^\bullet)^M} \mathbb{E}_{\boldsymbol{\sigma}} \left[ \sup_{\|\boldsymbol{u}\|_2 \leq 1} \frac{1}{M} \sum_{m=1}^M \sigma_m \lambda \operatorname{Tr}\left( \boldsymbol{u} \boldsymbol{u}^\top \boldsymbol{E}_{i_{k,m}^\bullet, j_{k,m}^\bullet}^{\mathcal{Z}} \right) \right]$$

$$= \frac{\lambda}{M} \mathbb{E}_{((i_{k,m}^\bullet, j_{k,m}^\bullet))_{m=1}^M \sim (\mathcal{D}_k^\bullet)^M} \mathbb{E}_{\boldsymbol{\sigma}} \left[ \sup_{\|\boldsymbol{u}\|_2 \leq 1} \sum_{m=1}^M \sigma_m \operatorname{Tr}\left( \boldsymbol{u}^\top \boldsymbol{E}_{i_{k,m}^\bullet, j_{k,m}^\bullet}^{\mathcal{Z}} \boldsymbol{u} \right) \right]$$

$$= \frac{\lambda}{M} \mathbb{E}_{((i_{k,m}^\bullet, j_{k,m}^\bullet))_{m=1}^M \sim (\mathcal{D}_k^\bullet)^M} \mathbb{E}_{\boldsymbol{\sigma}} \left[ \sup_{\|\boldsymbol{u}\|_2 \leq 1} \operatorname{Tr}\left( \boldsymbol{u}^\top \left( \sum_{m=1}^M \sigma_m \boldsymbol{E}_{i_{k,m}^\bullet, j_{k,m}^\bullet}^{\mathcal{Z}} \right) \boldsymbol{u} \right) \right]$$

$$= \frac{\lambda}{M} \mathbb{E}_{((i_{k,m}^\bullet, j_{k,m}^\bullet))_{m=1}^M \sim (\mathcal{D}_k^\bullet)^M} \mathbb{E}_{\boldsymbol{\sigma}} \left[ \left\| \sum_{m=1}^M \sigma_m \boldsymbol{E}_{i_{k,m}^\bullet, j_{k,m}^\bullet}^{\mathcal{Z}} \right\|_{\mathrm{op},2} \right], \tag{58}$$

where $\|\cdot\|_{\mathrm{op},2}$ denotes the operator norm with respect to the 2-norm defined by

$$\|\boldsymbol{A}\|_{\mathrm{op},2} := \max_{\|\boldsymbol{u}\|_2 \leq 1} \|\boldsymbol{A} \boldsymbol{u}\|_2 \tag{59}$$

To evaluate this, we can apply the following the matrix Bernstein inequality.

**Theorem 3** ([41] Theorem 6.6.1)**.** *Let $\boldsymbol{A}_1, \boldsymbol{A}_2, \ldots, \boldsymbol{A}_M \in \operatorname{Sym}^{|\mathcal{V}|}$ be independent random matrices that satisfies*

$$\mathbb{E} \boldsymbol{A}_m = \boldsymbol{O}, \quad \|\boldsymbol{A}_m\|_{\mathrm{op},2} \leq \sigma. \tag{60}$$

*Then*

$$\mathbb{E} \left\| \sum_{m=1}^M \boldsymbol{A}_m \right\|_{\mathrm{op},2} \leq \sqrt{2v\left( \sum_{m=1}^M \boldsymbol{A}_m \right) \ln |\mathcal{V}|} + \frac{1}{3}\sigma \ln |\mathcal{V}|, \tag{61}$$

*where $v$ is the matrix variance statistics defined by*

$$v(\boldsymbol{A}) := \left\| \mathbb{E} \boldsymbol{A}^2 \right\|_{\mathrm{op},2}. \tag{62}$$

Note that $v\left( \sum_{m=1}^M \boldsymbol{A}_m \right) = \left\| \sum_{m=1}^M \mathbb{E} \boldsymbol{A}_m^2 \right\|_{\mathrm{op},2} = \sum_{m=1}^M \left\| \mathbb{E} \boldsymbol{A}_m^2 \right\|_{\mathrm{op},2}$ is valid since $\boldsymbol{A}_1, \boldsymbol{A}_2, \ldots, \boldsymbol{A}_M \in \operatorname{Sym}^{|\mathcal{V}|}$ are independent. We apply Theorem 3 to the right hand side of (58) by substituting $\boldsymbol{A}_m$ by $\sigma_m \boldsymbol{E}_{i_{k,m}^\bullet, j_{k,m}^\bullet}^{\mathcal{Z}}$. Here, $\mathbb{E} \sigma_m \boldsymbol{E}_{i_{k,m}^\bullet, j_{k,m}^\bullet}^{\mathcal{Z}} = \boldsymbol{O}$ is valid because $\mathbb{E} \sigma_m = 0$. The singular values of $\sigma_m \boldsymbol{E}_{i_{k,m}^\bullet, j_{k,m}^\bullet}^{\mathcal{Z}}$ is equal to those of $\sigma_m \tilde{\boldsymbol{E}}^{\mathcal{Z}}$, where

$$\tilde{\boldsymbol{E}}^{\mathbb{R}^D} := \begin{bmatrix} +1 & -1 \\ -1 & +1 \end{bmatrix}, \qquad \tilde{\boldsymbol{E}}^{\mathbb{L}^D} := \begin{bmatrix} 0 & -\frac{1}{2} \\ -\frac{1}{2} & 0 \end{bmatrix}, \qquad \tilde{\boldsymbol{E}}^{\mathbb{S}^D} = \tilde{\boldsymbol{E}}^{\mathbb{I}^D} := \begin{bmatrix} 0 & +\frac{1}{2} \\ +\frac{1}{2} & 0 \end{bmatrix}. \tag{63}$$

Those singular values equal to the square root of eigenvalues of $\left( \sigma_m \tilde{\boldsymbol{E}}^{\mathcal{Z}} \right)^2 = \left( \tilde{\boldsymbol{E}}^{\mathcal{Z}} \right)^2$. Here,

$$\left( \tilde{\boldsymbol{E}}^{\mathbb{R}^D} \right)^2 = \begin{bmatrix} +2 & -2 \\ -2 & +2 \end{bmatrix}, \qquad \left( \tilde{\boldsymbol{E}}^{\mathbb{L}^D} \right)^2 = \left( \tilde{\boldsymbol{E}}^{\mathbb{S}^D} \right)^2 = \left( \tilde{\boldsymbol{E}}^{\mathbb{I}^D} \right)^2 := \begin{bmatrix} +\frac{1}{4} & 0 \\ 0 & +\frac{1}{4} \end{bmatrix}, \tag{64}$$

and the eigenvalues of $\left(\tilde{\boldsymbol{E}}^{\mathbb{R}^D}\right)^2$ are $0, +4$ and those of $\left(\tilde{\boldsymbol{E}}^{\mathbb{L}^D}\right)^2$, $\left(\tilde{\boldsymbol{E}}^{\mathbb{S}^D}\right)^2$ and $\left(\tilde{\boldsymbol{E}}^{\mathbb{I}^D}\right)^2$ are $+\frac{1}{4}, +\frac{1}{4}$. Hence, the singular values of $\tilde{\boldsymbol{E}}^{\mathbb{R}^D}$ are $0, +2$ and those of $\tilde{\boldsymbol{E}}^{\mathbb{L}^D}$, $\tilde{\boldsymbol{E}}^{\mathbb{S}^D}$ and $\tilde{\boldsymbol{E}}^{\mathbb{I}^D}$ are $+\frac{1}{2}, +\frac{1}{2}$.

Hence, we have that $\left\|\sigma_m \boldsymbol{E}^{\mathbb{R}^D}_{i^\bullet_{k,m}, j^\bullet_{k,m}}\right\|_{\text{op},2} = +2$ and $\left\|\sigma_m \boldsymbol{E}^{\mathbb{L}^D}_{i^\bullet_{k,m}, j^\bullet_{k,m}}\right\|_{\text{op},2} = \left\|\sigma_m \boldsymbol{E}^{\mathbb{S}^D}_{i^\bullet_{k,m}, j^\bullet_{k,m}}\right\|_{\text{op},2} = \left\|\sigma_m \boldsymbol{E}^{\mathbb{I}^D}_{i^\bullet_{k,m}, j^\bullet_{k,m}}\right\|_{\text{op},2} = +\frac{1}{2}$. We define $\kappa_{\mathcal{Z}} := \left\|\sigma_m \boldsymbol{E}^{\mathcal{Z}}_{i^\bullet_{k,m}, j^\bullet_{k,m}}\right\|_{\text{op},2}$. The remaining part is the evaluation of $v\left(\sum_{m=1}^M \sigma_m \boldsymbol{E}^{\mathcal{Z}}_{i^\bullet_{k,m}, j^\bullet_{k,m}}\right)$.

We have that

$$v\left(\sum_{m=1}^M \sigma_m \boldsymbol{E}^{\mathcal{Z}}_{i^\bullet_{k,m}, j^\bullet_{k,m}}\right) = \sum_{m=1}^M \left\|\mathbb{E}\left(\sigma_m \boldsymbol{E}^{\mathcal{Z}}_{i^\bullet_{k,m}, j^\bullet_{k,m}}\right)^2\right\|_{\text{op},2} = M\left\|\mathbb{E}\left(\boldsymbol{E}^{\mathcal{Z}}_{i^\bullet_{k,m}, j^\bullet_{k,m}}\right)^2\right\|_{\text{op},2} = M\nu^\bullet_k. \tag{65}$$

Therefore, we have that

$$\mathbb{E}_{\left((i^\bullet_{k,m}, j^\bullet_{k,m})\right)_{m=1}^M \sim \left(\mathcal{D}^\bullet_k\right)^M} \mathbb{E}_{\boldsymbol{\sigma}}\left[\left\|\sum_{m=1}^M \sigma_m \boldsymbol{E}^{\mathcal{Z}}_{i^\bullet_{k,m}, j^\bullet_{k,m}}\right\|_{\text{op},2}\right] \le \sqrt{2M\nu^\bullet_k \ln|\mathcal{V}|} + \frac{\kappa_{\mathcal{Z}}}{3}\ln|\mathcal{V}|, \tag{66}$$

which completes the proof. $\qquad\square$

*Proof of Lemma 3.* For $((x_{1,m}, x_{2,m}, \ldots, x_{K,m}))_{m=1}^M \in (\mathcal{X}_1 \times \mathcal{X}_2 \times \cdots \times \mathcal{X}_K)^M$ and $\sigma_2, \sigma_3, \ldots, \sigma_M \in \{-1, +1\}$, for all $\epsilon \in \mathbb{R}_{>0}$, there exist $\left(f^+_{1,1}, f^+_{2,1}, \ldots, f^+_{K,1}\right), \left(f^-_{1,1}, f^-_{2,1}, \ldots, f^-_{K,1}\right) \in \mathcal{F}$ such that

$$\begin{aligned} \sup_{(f_{1,1}, f_{2,1}, \ldots, f_{K,1}) \in \mathcal{F}} &[l(f_1(x_{1,1}), f_{2,1}(x_{2,1}), \ldots, f_{K,1}(x_{K,1})) + u_1(f_{1,1}, f_{2,1}, \ldots, f_{K,1})] \\ &< l\left(f^+_{1,1}(x_{1,1}), f^+_{2,1}(x_{2,1}), \ldots, f^+_{K,1}(x_{K,1})\right) + u_1\left(f^+_{1,1}, f^+_{2,1}, \ldots, f^+_{K,1}\right) + \epsilon, \end{aligned} \tag{67}$$

and

$$\begin{aligned} \sup_{(f_{1,1}, f_{2,1}, \ldots, f_{K,1}) \in \mathcal{F}} &[-l(f_1(x_{1,1}), f_{2,1}(x_{2,1}), \ldots, f_{K,1}(x_{K,1})) + u_1(f_{1,1}, f_{2,1}, \ldots, f_{K,1})] \\ &< -l\left(f^-_{1,1}(x_{1,1}), f^-_{2,1}(x_{2,1}), \ldots, f^-_{K,1}(x_{K,1})\right) + u_1\left(f^-_{1,1}, f^-_{2,1}, \ldots, f^-_{K,1}\right) + \epsilon, \end{aligned} \tag{68}$$

respectively, where $u_1(f_1, f_2, \ldots, f_K) := \sum_{m=2}^{M} \sigma_m l(f_1(x_{1,m}), f_2(x_{2,m}), \ldots, f_K(x_{K,m}))$. Then, we have that

$$\mathbb{E}_{\sigma_1} \sup_{(f_{1,1}, f_{2,1}, \ldots, f_{K,1}) \in \mathcal{F}} [\sigma_1 l(f_{1,1}(x_{1,1}), f_{2,1}(x_{2,1}), \ldots, f_{K,1}(x_{K,1})) + u_1(f_{1,1}, f_{2,1}, \ldots, f_{K,1})]$$

$$= \frac{1}{2} \Bigg\{ \sup_{(f_{1,1}, f_{2,1}, \ldots, f_{K,1}) \in \mathcal{F}} [l(f_{1,1}(x_{1,1}), f_{2,1}(x_{2,1}), \ldots, f_{K,1}(x_{K,1})) + u_1(f_{1,1}, f_{2,1}, \ldots, f_{K,1})]$$

$$+ \sup_{(f_{1,1}, f_{2,1}, \ldots, f_{K,1}) \in \mathcal{F}} [-l(f_{1,1}(x_{1,1}), f_{2,1}(x_{2,1}), \ldots, f_{K,1}(x_{K,1})) + u_1(f_{1,1}, f_{2,1}, \ldots, f_{K,1})] \Bigg\}$$

$$< \frac{1}{2} \Bigg\{ \Big[ l\Big(f_{1,1}^+(x_{1,1}), f_{2,1}^+(x_{2,1}), \ldots, f_{K,1}^+(x_{K,1})\Big) + u_1\Big(f_{1,1}^+, f_{2,1}^+, \ldots, f_{K,1}^+\Big) \Big]$$

$$+ \Big[ -l\Big(f_{1,1}^-(x_{1,1}), f_{2,1}^-(x_{2,1}), \ldots, f_{K,1}^-(x_{K,1})\Big) + u_1\Big(f_{1,1}^-, f_{2,1}^-, \ldots, f_{K,1}^-\Big) \Big] \Bigg\} + \epsilon$$

$$= \frac{1}{2} \Bigg\{ \Big[ l\Big(f_{1,1}^+(x_{1,1}), f_{2,1}^+(x_{2,1}), \ldots, f_{K,1}^+(x_{K,1})\Big) - l\Big(f_{1,1}^-(x_{1,1}), f_{2,1}^-(x_{2,1}), \ldots, f_{K,1}^-(x_{K,1})\Big) \Big]$$

$$+ \Big[ u_1\Big(f_{1,1}^+, f_{2,1}^+, \ldots, f_{K,1}^+\Big) + u_1\Big(f_{1,1}^-, f_{2,1}^-, \ldots, f_{K,1}^-\Big) \Big] \Bigg\} + \epsilon$$

$$= \frac{1}{2} \Bigg\{ \Big[ l\Big(f_{1,1}^+(x_{1,1}), f_{2,1}^+(x_{2,1}), \ldots, f_{K,1}^+(x_{K,1})\Big) - l\Big(f_{1,1}^-(x_{1,1}), f_{2,1}^+(x_{2,1}), \ldots, f_{K,1}^+(x_{K,1})\Big)$$

$$+ l\Big(f_{1,1}^-(x_{1,1}), f_{2,1}^+(x_{2,1}), \ldots, f_{K,1}^+(x_{K,1})\Big) - l\Big(f_{1,1}^-(x_{1,1}), f_{2,1}^-(x_{2,1}), \ldots, f_{K,1}^+(x_{K,1})\Big)$$

$$\ldots$$

$$+ l\Big(f_{1,1}^-(x_{1,1}), f_{2,1}^-(x_{2,1}), \ldots, f_{K,1}^+(x_{K,1})\Big) - l\Big(f_{1,1}^-(x_{1,1}), f_{2,1}^-(x_{2,1}), \ldots, f_{K,1}^-(x_{K,1})\Big) \Big]$$

$$+ \Big[ u_1\Big(f_{1,1}^+, f_{2,1}^+, \ldots, f_{K,1}^+\Big) + u_1\Big(f_{1,1}^-, f_{2,1}^-, \ldots, f_{K,1}^-\Big) \Big] \Bigg\} + \epsilon$$

$$= \frac{1}{2} \Bigg\{ \sum_{k=1}^{K} e_{k,1} L_k \Big( f_{k,1}^+(x_{k,1}) - f_{k,1}^-(x_{k,1}) \Big)$$

$$+ \Big[ u_1\Big(f_{1,1}^+, f_{2,1}^+, \ldots, f_{K,1}^+\Big) + u_1\Big(f_{1,1}^-, f_{2,1}^-, \ldots, f_{K,1}^-\Big) \Big] \Bigg\} + \epsilon$$

$$= \frac{1}{2} \Bigg\{ \Big[ (+1) \sum_{k=1}^{K} e_{k,1} L_k f_{k,1}^+(x_{k,1}) + u_1\Big(f_{1,1}^+, f_{2,1}^+, \ldots, f_{K,1}^+\Big) \Big]$$

$$+ \Big[ (-1) \sum_{k=1}^{K} e_{k,1} L_k f_{k,1}^-(x_{k,1}) + u_1\Big(f_{1,1}^-, f_{2,1}^-, \ldots, f_{K,1}^-\Big) \Big] \Bigg\} + \epsilon$$

$$= \mathbb{E}_{\sigma_1} \sup_{(f_{1,1}, f_{2,1}, \ldots, f_{K,1}) \in \mathcal{F}} \Big[ \sigma_1 \sum_{k=1}^{K} e_{k,1} L_k f_{k,1}(x_{k,1}) + u_1(f_{1,1}, f_{2,1}, \ldots, f_{K,1}) \Big] + \epsilon,$$

(69)

where $e_{k,1} \in \{-1.0, +1\}$ is defined by

$$e_{k,1} := \operatorname{sgn}\Bigg( l\Big(f_{1,1}^+(x_{1,1}), \ldots, f_{k-1,1}^+(x_{k-1,1}), f_{k,1}^+(x_{k,1}), f_{k+1,1}^-(x_{k+1,1}), \ldots, f_{K,1}^-(x_{K,1})\Big)$$

$$- l\Big(f_{1,1}^+(x_{1,1}), \ldots, f_{k-1,1}^+(x_{k-1,1}), f_{k,1}^-(x_{k,1}), f_{k+1,1}^-(x_{k+1,1}), \ldots, f_{K,1}^-(x_{K,1})\Big) \Bigg)$$

(70)

Since the above is valid for all $\epsilon \in \mathbb{R}_{>0}$, we have that

$$\mathbb{E}_{\sigma_1} \sup_{(f_{1,1}, f_{2,1}, \ldots, f_{K,1}) \in \mathcal{F}} [\sigma_1 l(f_{1,1}(x_{1,1}), f_{2,1}(x_{2,1}), \ldots, f_{K,1}(x_{K,1})) + u_1(f_{1,1}, f_{2,1}, \ldots, f_{K,1})]$$

$$< \mathbb{E}_{\sigma_1} \sup_{(f_{1,1}, f_{2,1}, \ldots, f_{K,1}) \in \mathcal{F}} \left[ \sigma_1 \sum_{k=1}^{K} e_{k,1} L_k f_{k,1}(x_{k,1}) + u_1(f_{1,1}, f_{2,1}, \ldots, f_{K,1}) \right].$$

$$(71)$$

Taking expectation with respect to $\sigma_2, \sigma_3, \ldots, \sigma_M$, we have that

$$\mathbb{E}_{\sigma_1, \sigma_2, \ldots, \sigma_M} \sup_{(f_{1,1}, f_{2,1}, \ldots, f_{K,1}) \in \mathcal{F}} [\sigma_1 l(f_{1,1}(x_{1,1}), f_{2,1}(x_{2,1}), \ldots, f_{K,1}(x_{K,1})) + u_1(f_{1,1}, f_{2,1}, \ldots, f_{K,1})]$$

$$< \mathbb{E}_{\sigma_1, \sigma_2, \ldots, \sigma_M} \sup_{(f_{1,1}, f_{2,1}, \ldots, f_{K,1}) \in \mathcal{F}} \left[ \sigma_1 \sum_{k=1}^{K} e_{k,1} L_k f_{k,1}(x_{k,1}) + u_1(f_{1,1}, f_{2,1}, \ldots, f_{K,1}) \right].$$

$$(72)$$

Likewise, if we fix $\sigma_1, \sigma_2, \ldots, \sigma_{m'-1}, \sigma_{m'+1}, \sigma_{m'+2}, \ldots, \sigma_M$ we have that

$$\mathbb{E}_{\sigma_{m'}} \sup_{\left(f_{1,m'}, f_{2,m'}, \ldots, f_{K,m'}\right) \in \mathcal{F}} \left[ \sigma_{m'} l(f_{1,m'}(x_{1,m'}), f_{2,m'}(x_{2,m'}), \ldots, f_{K,m'}(x_{K,m'})) \right.$$

$$\left. + u_{m'}(f_{1,m'}, f_{2,m'}, \ldots, f_{K,m'}) \right]$$

$$\leq \mathbb{E}_{\sigma_{m'}} \sup_{\left(f_{1,m'}, f_{2,m'}, \ldots, f_{K,m'}\right) \in \mathcal{F}} \left[ \sigma_{m'} \sum_{k=1}^{K} e_{k,m'} L_k f_{k,m'}(x_{k,m'}) + u_{m'}(f_{1,m'}, f_{2,m'}, \ldots, f_{K,m'}) \right],$$

$$(73)$$

where

$$u_{m'}(f_1, f_2, \ldots, f_K) := \sum_{m=m'+1}^{m'-1} \sigma_{m'} \sum_{k=1}^{K} e_{k,m'} L_k f_{k,m'}(x_{k,m'}) + \sum_{m=m'+1}^{M} \sigma_m l(f_1(x_{1,m}), f_2(x_{2,m}), \ldots, f_K(x_{K,m}))$$

$$(74)$$

and $\left(f_{1,m'}^+, f_{2,m'}^+, \ldots, f_{K,m'}^+\right), \left(f_{1,m'}^-, f_{2,m'}^-, \ldots, f_{K,m'}^-\right) \in \mathcal{F}$ are functions that satisfies

$$\sup_{(f_{1,m'}, f_{2,m'}, \ldots, f_{K,m'}) \in \mathcal{F}} [l(f_{1,m'}(x_{1,m'}), f_{2,m'}(x_{2,m'}), \ldots, f_{K,m'}(x_{K,m'})) + u_{m'}(f_{1,m'}, f_{2,m'}, \ldots, f_{K,m'})]$$

$$< l\left(f_{1,m'}^+(x_{1,m'}), f_{2,m'}^+(x_{2,m'}), \ldots, f_{K,m'}^+(x_{K,m'})\right) + u_{m'}\left(f_{1,m'}^+, f_{2,m'}^+, \ldots, f_{K,m'}^+\right) + \epsilon,$$

$$(75)$$

and

$$\sup_{(f_{1,m'}, f_{2,m'}, \ldots, f_{K,m'}) \in \mathcal{F}} [-l(f_{1,m'}(x_{1,m'}), f_{2,m'}(x_{2,m'}), \ldots, f_{K,m'}(x_{K,m'})) + u_{m'}(f_{1,m'}, f_{2,m'}, \ldots, f_{K,m'})]$$

$$< -l\left(f_{1,m'}^-(x_{1,m'}), f_{2,m'}^-(x_{2,m'}), \ldots, f_{K,m'}^-(x_{K,m'})\right) + u_{m'}\left(f_{1,m'}^-, f_{2,m'}^-, \ldots, f_{K,m'}^-\right) + \epsilon,$$

$$(76)$$

respectively, and $e_{k,m'} \in \{-1.0, +1\}$ is defined by

$$e_{k,m'} := \text{sgn}\left( l\left(f_{1,m'}^+(x_{1,m'}), \ldots, f_{k-1,m'}^+(x_{k-1,m'}), f_{k,m'}^+(x_{k,m'}), f_{k+1,m'}^-(x_{k+1,m'}), \ldots, f_{K,m'}^-(x_{K,m'})\right) \right.$$

$$\left. - l\left(f_{1,m'}^+(x_{1,m'}), \ldots, f_{k-1,m'}^+(x_{k-1,m'}), f_{k,m'}^-(x_{k,m'}), f_{k+1,m'}^-(x_{k+1,m'}), \ldots, f_{K,1}^-(x_{K,1})\right) \right).$$

$$(77)$$

Taking expectation with respect to $\sigma_1, \sigma_2, \ldots, \sigma_{m'-1}, \sigma_{m'+1}, \sigma_{m'+2}, \ldots, \sigma_M$, we have that

$$
\mathbb{E}_{\sigma_1, \sigma_2, \ldots, \sigma_M} \sup_{(f_{1,m'}, f_{2,m'}, \ldots, f_{K,m'}) \in \mathcal{F}} \Big[ \sigma_{m'} l(f_{1,m'}(x_{1,m'}), f_{2,m'}(x_{2,m'}), \ldots, f_{K,m'}(x_{K,m'}))
$$
$$
+ u_{m'}(f_{1,m'}, f_{2,m'}, \ldots, f_{K,m'}) \Big]
$$
$$
\leq \mathbb{E}_{\sigma_1, \sigma_2, \ldots, \sigma_M} \sup_{(f_{1,m'}, f_{2,m'}, \ldots, f_{K,m'}) \in \mathcal{F}} \Big[ \sigma_{m'} \sum_{k=1}^{K} e_{k,m'} L_k f_{k,m'}(x_{k,m'}) + u_{m'}(f_{1,m'}, f_{2,m'}, \ldots, f_{K,m'}) \Big].
$$
(78)

By induction, we obtain that

$$
\mathbb{E}_{\sigma_1, \sigma_2, \ldots, \sigma_M} \sup_{(f_1, f_2, \ldots, f_K) \in \mathcal{F}} \Big[ \sum_{m=1}^{M} \sigma_m l(f_1(x_{1,m}), f_2(x_{2,m}), \ldots, f_K(x_{K,m})) \Big]
$$
$$
\leq \mathbb{E}_{\sigma_1, \sigma_2, \ldots, \sigma_M} \sup_{(f_1, f_2, \ldots, f_K) \in \mathcal{F}} \Big[ \sum_{m=1}^{M} \sigma_m \sum_{k=1}^{K} e_{k,m} L_k f_k(x_{k,m}) \Big]
$$
$$
= \mathbb{E}_{\sigma_1, \sigma_2, \ldots, \sigma_M} \sup_{(f_1, f_2, \ldots, f_K) \in \mathcal{F}} \Big[ \sum_{k=1}^{K} \sum_{m=1}^{M} \sigma_m e_{k,m} L_k f_k(x_{k,m}) \Big]
$$
$$
\leq \sum_{k=1}^{K} L_k \mathbb{E}_{\sigma_1, \sigma_2, \ldots, \sigma_M} \sup_{(f_1, f_2, \ldots, f_K) \in \mathcal{F}} \Big[ \sum_{m=1}^{M} \sigma_m e_{k,m} f_k(x_{k,m}) \Big]
$$
(79)
$$
= \sum_{k=1}^{K} L_k \mathbb{E}_{\sigma_1, \sigma_2, \ldots, \sigma_M} \sup_{f_k \in \mathcal{F}_k} \Big[ \sum_{m=1}^{M} \sigma_m e_{k,m} f_k(x_{k,m}) \Big]
$$
$$
= \sum_{k=1}^{K} L_k \mathbb{E}_{\sigma_1, \sigma_2, \ldots, \sigma_M} \sup_{f_k \in \mathcal{F}_k} \Big[ \sum_{m=1}^{M} \sigma_m f_k(x_{k,m}) \Big],
$$

which completes the proof. Here, the last equality holds since the distribution of the random variable $\sigma_m e_{k,m}$ is the same as that of $\sigma_m$.

$\square$

## C  Proof of Proposition 1

First, we recall the proposition.

**Proposition** (Proposition 1). *(a) Suppose that the data distribution is given by (6) and (7). Then we have that*

$$
\nu_{k,\mathcal{Z}}^{+} = \frac{1}{2} \cdot \frac{(1-r^+)\max_{i \in \mathcal{V}} \deg(i) + r^+(|\mathcal{V}|-1)}{[(1-r^+)|\mathcal{E}| + r^+|\mathcal{V}|(|\mathcal{V}|-1)]} \leq \frac{1}{2} \cdot \frac{\max_{i \in \mathcal{V}} \deg(i)}{|\mathcal{E}|},
$$
(80)
$$
\nu_{k,\mathcal{Z}}^{-} = \frac{1}{2} \cdot \frac{(1-r^-)[(|\mathcal{V}|-1) - \min_{i \in \mathcal{V}} \deg(i)] + r^-(|\mathcal{V}|-1)}{[(1-r^-)[|\mathcal{V}|(|\mathcal{V}|-1) - |\mathcal{E}|] + r^-|\mathcal{V}|(|\mathcal{V}|-1)]} \leq \frac{1}{2} \cdot \frac{(|\mathcal{V}|-1) - \min_{i \in \mathcal{V}} \deg(i)}{|\mathcal{V}|(|\mathcal{V}|-1) - |\mathcal{E}|},
$$
(81)

*for $\mathcal{Z} = \mathbb{L}^D, \mathbb{I}^D, \mathbb{S}^D$, and for $\mathcal{Z} = \mathbb{R}^D$ we have that*

$$
\nu_{k,\mathbb{R}^D}^{+} \leq \frac{2[(1-r^+)\{2\max_{i \in \mathcal{V}} \deg(i)\} + r^+|\mathcal{V}|]}{(1-r^+)|\mathcal{E}| + r^+|\mathcal{V}|(|\mathcal{V}|-1)} \leq 4 \cdot \frac{\max_{i \in \mathcal{V}} \deg(i)}{|\mathcal{E}|},
$$
(82)
$$
\nu_{k,\mathbb{R}^D}^{-} \leq 4 \frac{1}{|\mathcal{V}|} \frac{(|\mathcal{V}|-1) - (1-r^-)\min_{i \in \mathcal{V}} \deg(i)}{(|\mathcal{V}|-1) - (1-r^-)\frac{|\mathcal{E}|}{|\mathcal{V}|}} \leq 4 \cdot \frac{1}{|\mathcal{V}|} \frac{(|\mathcal{V}|-1) - \min_{i \in \mathcal{V}} \deg(i)}{(|\mathcal{V}|-1) - \frac{|\mathcal{E}|}{|\mathcal{V}|}}.
$$
(83)

*(b) If the conditional distribution of negative pairs is given by (8), then we have that*

$$
\nu_{k,\mathcal{Z}}^{-} = \frac{1}{2} \cdot \frac{(1-r^-)\max_{i \in \mathcal{V}} \deg(i) + r^-(|\mathcal{V}|-1)}{[(1-r^-)|\mathcal{E}| + r^-|\mathcal{V}|(|\mathcal{V}|-1)]} \leq \frac{1}{2} \cdot \frac{\max_{i \in \mathcal{V}} \deg(i)}{|\mathcal{E}|},
$$
(84)

*for $\mathcal{Z} = \mathbb{L}^D, \mathbb{I}^D, \mathbb{S}^D$, and*

$$\nu_{k,\mathbb{R}^D}^- = 2\left[\frac{(1-r^-)\max_{i\in\mathcal{V}}\deg(i) + r^-(|\mathcal{V}|-1)}{(1-r^-)|\mathcal{E}| + r^-|\mathcal{V}|(|\mathcal{V}|-1)} + 1\right] \leq 2\left[\frac{\max_{i\in\mathcal{V}}\deg(i)}{|\mathcal{E}|} + 1\right]. \quad (85)$$

*Proof of Proposition 1.* **(a)** Since $\mathbb{P}\left[\left(i_{k,m}^+, j_{k,m}^+\right)\right]$ is given by (6), the $i$-th row and the $j$-th column of $\mathbb{E}_{\left(i_{k,m}^\bullet, j_{k,m}^\bullet\right)} \boldsymbol{E}_{i_{k,m}^\bullet, j_{k,m}^\bullet}^{\mathcal{Z}}{}^\top \boldsymbol{E}_{i_{k,m}^\bullet, j_{k,m}^\bullet}^{\mathcal{Z}}$ is given by

$$\left[\mathbb{E}_{\left(i_{k,m}^+, j_{k,m}^+\right)} \boldsymbol{E}_{i_{k,m}^+, j_{k,m}^+}^{\mathcal{Z}}{}^\top \boldsymbol{E}_{i_{k,m}^+, j_{k,m}^+}^{\mathcal{Z}}\right]_{i,j} = \begin{cases} \frac{1}{2} \cdot \frac{(1-r^+)\deg(i) + r^+(|\mathcal{V}|-1)}{[(1-r^+)|\mathcal{E}| + r^+|\mathcal{V}|(|\mathcal{V}|-1)]} & \text{if } i = j, \\ 0 & \text{if } i \neq j, \end{cases} \quad (86)$$

*for $\mathcal{Z} = \mathbb{L}^D, \mathbb{S}^D, \mathbb{I}^D$, and*

$$\left[\mathbb{E}_{\left(i_{k,m}^+, j_{k,m}^+\right)} \boldsymbol{E}_{i_{k,m}^+, j_{k,m}^+}^{\mathcal{Z}}{}^\top \boldsymbol{E}_{i_{k,m}^+, j_{k,m}^+}^{\mathcal{Z}}\right]_{i,j} = \begin{cases} 2[(1-r^+)\deg(i) + r^+(|\mathcal{V}|-1)]p^+ & \text{if } i = j, \\ -2p^+ & \text{if } i \neq j \text{ and } (i,j) \in \mathcal{E}, \\ -2r^+ p^+ & \text{if } i \neq j \text{ and } (i,j) \notin \mathcal{E}, \end{cases} \quad (87)$$

*for $\mathcal{Z} = \mathbb{R}^D$, where $p^+$ is given by $p^+ = \frac{1}{(1-r^+)|\mathcal{E}| + r^+|\mathcal{V}|(|\mathcal{V}|-1)}$. In other words, we have that*

$$\begin{aligned} &\mathbb{E}_{\left(i_{k,m}^+, j_{k,m}^+\right)} \boldsymbol{E}_{i_{k,m}^+, j_{k,m}^+}^{\mathcal{Z}}{}^\top \boldsymbol{E}_{i_{k,m}^+, j_{k,m}^+}^{\mathcal{Z}} \\ &= 2\left[\left\{(1-r^+)\operatorname{diag}(\boldsymbol{d}) + r^+(|\mathcal{V}|-1)\mathbf{I}\right\} - \left\{(1-r^+)\boldsymbol{A} + r^+(\mathbf{1}\mathbf{1}^\top - \mathbf{I})\right\}\right]p^+ \\ &= 2\left[(1-r^+)\boldsymbol{L} + r^+(|\mathcal{V}|\mathbf{I} - \mathbf{1}\mathbf{1}^\top)\right]p^+, \end{aligned} \quad (88)$$

*where $\operatorname{diag}(\boldsymbol{d}) \in \mathbb{R}^{|\mathcal{V}|,|\mathcal{V}|}$ is given by*

$$\operatorname{diag}(\boldsymbol{d}) := \begin{bmatrix} \deg(1) & & & \\ & \deg(2) & & \\ & & \ddots & \\ & & & \deg(|\mathcal{V}|) \end{bmatrix}, \quad (89)$$

$\boldsymbol{A} \in \mathbb{R}^{|\mathcal{V}|,|\mathcal{V}|}$ is the adjacency matrix defined by

$$[\boldsymbol{A}]_{i,j} := \begin{cases} 1 & \text{if } (i,j) \in \mathcal{E}, \\ 0 & \text{if } (i,j) \notin \mathcal{E}, \end{cases} \quad (90)$$

and $\boldsymbol{L} \in \mathbb{R}^{|\mathcal{V}|,|\mathcal{V}|}$ is given by $\boldsymbol{L} := \operatorname{diag}(\boldsymbol{d}) - \boldsymbol{A}$.

Hence, we have that

$$\begin{aligned} \nu_{k,\mathcal{Z}}^+ &= \frac{1}{2} \cdot \frac{(1-r^+)\max_{i\in\mathcal{V}}\deg(i) + r^+(|\mathcal{V}|-1)}{(1-r^+)|\mathcal{E}| + r^+|\mathcal{V}|(|\mathcal{V}|-1)} \\ &= \frac{1}{2} \cdot \frac{\max_{i\in\mathcal{V}}\deg(i)}{|\mathcal{E}|} \cdot \frac{(1-r^+) + r^+\frac{1}{\max_{i\in\mathcal{V}}\deg(i)}(|\mathcal{V}|-1)}{(1-r^+) + r^+\frac{|\mathcal{V}|}{|\mathcal{E}|}(|\mathcal{V}|-1)}, \end{aligned} \quad (91)$$

for $\mathcal{Z} = \mathbb{L}^D, \mathbb{S}^D, \mathbb{I}^D$. Here, since $\max_{i\in\mathcal{V}}\deg(i) \geq \frac{|\mathcal{E}|}{|\mathcal{V}|}$, the right hand side is decreasing with respect to $r^+$ and takes its maximum $\frac{1}{2} \cdot \frac{\max_{i\in\mathcal{V}}\deg(i)}{|\mathcal{E}|}$ if $r^+ = 0$. Also, for $\mathcal{Z} = \mathbb{R}^D$, we have that

$$\begin{aligned} \nu_{k,\mathcal{Z}}^+ &= \left\|2\left[(1-r^+)\boldsymbol{L} + r^+(|\mathcal{V}|\mathbf{I} - \mathbf{1}\mathbf{1}^\top)\right]p^+\right\|_{\text{op},2} \\ &\leq 2\left[(1-r^+)\|\boldsymbol{L}\|_{\text{op},2} + r^+\left\||\mathcal{V}|\mathbf{I} - \mathbf{1}\mathbf{1}^\top\right\|_{\text{op},2}\right]p^+ \\ &\leq 2\left[(1-r^+)\left\{2\max_{i\in\mathcal{V}}\deg(i)\right\} + r^+|\mathcal{V}|\right]p^+, \\ &= \frac{2[(1-r^+)\{2\max_{i\in\mathcal{V}}\deg(i)\} + r^+|\mathcal{V}|]}{(1-r^+)|\mathcal{E}| + r^+|\mathcal{V}|(|\mathcal{V}|-1)} \\ &= 4 \cdot \frac{\max_{i\in\mathcal{V}}\deg(i)}{|\mathcal{E}|} \cdot \frac{(1-r^+) + r^+|\mathcal{V}|\frac{1}{2\max_{i\in\mathcal{V}}\deg(i)}}{(1-r^+) + r^+|\mathcal{V}|\frac{|\mathcal{V}|-1}{|\mathcal{E}|}}, \end{aligned} \quad (92)$$

Here, since $2\max_{i\in\mathcal{V}}\deg\,(i)\geq 2\frac{|\mathcal{E}|}{|\mathcal{V}|}\geq\frac{|\mathcal{E}|}{|\mathcal{V}|-1}$, the right hand side is decreasing with respect to $r^+$ and takes its maximum $4\cdot\frac{\max_{i\in\mathcal{V}}\deg\,(i)}{|\mathcal{E}|}$ if $r^+=0$.

Similarly, we have that

$$\left[\mathbb{E}_{\left(i_{k,m}^-,j_{k,m}^-\right)}\boldsymbol{E}_{i_{k,m}^-,j_{k,m}^-}^{\mathcal{Z}}{}^\top\boldsymbol{E}_{i_{k,m}^-,j_{k,m}^-}^{\mathcal{Z}}\right]_{i,j}=\begin{cases}\frac{1}{2}\cdot\frac{\left(1-r^-\right)\left(\left(|\mathcal{V}|-1\right)-\deg\,(i)\right)+r^-\left(|\mathcal{V}|-1\right)}{\left(1-r^-\right)\left(|\mathcal{V}|(|\mathcal{V}|-1)-|\mathcal{E}|\right)+r^-|\mathcal{V}|(|\mathcal{V}|-1)}&\text{if }i=j,\\0&\text{if }i\neq j.\end{cases}\tag{93}$$

for $\mathcal{Z}=\mathbb{L}^D,\mathbb{S}^D,\mathbb{I}^D$, and

$$\begin{aligned}&\left[\mathbb{E}_{\left(i_{k,m}^-,j_{k,m}^-\right)}\boldsymbol{E}_{i_{k,m}^-,j_{k,m}^-}^{\mathcal{Z}}{}^\top\boldsymbol{E}_{i_{k,m}^-,j_{k,m}^-}^{\mathcal{Z}}\right]_{i,j}\\&=\begin{cases}2[(1-r^+)\{(|\mathcal{V}|-1)-\deg\,(i)\}+r^+(|\mathcal{V}|-1)]p^+&\text{if }i=j,\\-2r^+p^+&\text{if }i\neq j\text{ and }(i,j)\in\mathcal{E},\\-2p^+&\text{if }i\neq j\text{ and }(i,j)\notin\mathcal{E},\end{cases}\end{aligned}\tag{94}$$

for $\mathcal{Z}=\mathbb{R}^D$, where $p^-$ is given by $p^-=\frac{1}{(1-r^-)(|\mathcal{V}|(|\mathcal{V}|-1)-|\mathcal{E}|)+r^-|\mathcal{V}|(|\mathcal{V}|-1)}$. In other words, we have that

$$\begin{aligned}&\mathbb{E}_{\left(i_{k,m}^-,j_{k,m}^-\right)}\boldsymbol{E}_{i_{k,m}^-,j_{k,m}^-}^{\mathcal{Z}}{}^\top\boldsymbol{E}_{i_{k,m}^-,j_{k,m}^-}^{\mathcal{Z}}\\&=2\big[\big\{\big(1-r^-\big)((|\mathcal{V}|-1)\mathbf{I}-\mathrm{diag}\,(\boldsymbol{d}))+r^-(|\mathcal{V}|-1)\mathbf{I}\big\}-\big\{\big(1-r^-\big)\big(\mathbf{1}\mathbf{1}^\top-\mathbf{I}-\boldsymbol{A}\big)+r^-\big(\mathbf{1}\mathbf{1}^\top-\mathbf{I}\big)\big\}\big]p^-\\&=2\big[\big(1-r^-\big)\big\{((|\mathcal{V}|-1)\mathbf{I}-\mathrm{diag}\,(\boldsymbol{d}))-\big(\mathbf{1}\mathbf{1}^\top-\mathbf{I}-\boldsymbol{A}\big)\big\}+r^-\big(|\mathcal{V}|\mathbf{I}-\mathbf{1}\mathbf{1}^\top\big)\big]p^-,\end{aligned}\tag{95}$$

Hence, for $\mathcal{Z}=\mathbb{L}^D,\mathbb{S}^D,\mathbb{I}^D$, we have that

$$\begin{aligned}\nu_k^-&=\frac{1}{2}\cdot\frac{(1-r^-)((|\mathcal{V}|-1)-\min_{i\in\mathcal{V}}\deg\,(i))+r^-(|\mathcal{V}|-1)}{(1-r^-)(|\mathcal{V}|(|\mathcal{V}|-1)-|\mathcal{E}|)+r^-|\mathcal{V}|(|\mathcal{V}|-1)}\\&=\frac{1}{2}\cdot\frac{1}{|\mathcal{V}|}\frac{(|\mathcal{V}|-1)-(1-r^-)(\min_{i\in\mathcal{V}}\deg\,(i))}{(|\mathcal{V}|-1)-(1-r^-)\frac{|\mathcal{E}|}{|\mathcal{V}|}}\end{aligned}\tag{96}$$

Here, since $\min_{i\in\mathcal{V}}\deg\,(i)\leq\frac{|\mathcal{E}|}{|\mathcal{V}|}$, the right hand side is decreasing with respect to $r$ and takes its maximum $\frac{1}{2}\cdot\frac{1}{|\mathcal{V}|}\frac{(|\mathcal{V}|-1)-(\min_{i\in\mathcal{V}}\deg\,(i))}{(|\mathcal{V}|-1)-\frac{|\mathcal{E}|}{|\mathcal{V}|}}$ if $r=0$.

Also, for $\mathcal{Z}=\mathbb{R}^D$, we have that

$$\begin{aligned}\nu_{k,\mathcal{Z}}^-&=\big\|2\big[\big(1-r^-\big)\big\{((|\mathcal{V}|-1)\mathbf{I}-\mathrm{diag}\,(\boldsymbol{d}))-\big(\mathbf{1}\mathbf{1}^\top-\mathbf{I}-\boldsymbol{A}\big)\big\}+r^-\big(|\mathcal{V}|\mathbf{I}-\mathbf{1}\mathbf{1}^\top\big)\big]p^-\big\|_{\mathrm{op},2}\\&\leq 2\big[\big(1-r^-\big)\big\{\|(|\mathcal{V}|-1)\mathbf{I}-\mathrm{diag}\,(\boldsymbol{d})\|_{\mathrm{op},2}-\big\|\mathbf{1}\mathbf{1}^\top-\mathbf{I}-\boldsymbol{A}\big\|_{\mathrm{op},2}\big\}+r^-\big\||\mathcal{V}|\mathbf{I}-\mathbf{1}\mathbf{1}^\top\big\|_{\mathrm{op},2}\big]p^-\\&\leq 2\big[\big(1-r^+\big)2\big\{(|\mathcal{V}|-1)-\min_{i\in\mathcal{V}}\deg\,(i)\big\}+r^-|\mathcal{V}|\big]p^+,\\&=2\frac{(1-r^+)2\{(|\mathcal{V}|-1)-\min_{i\in\mathcal{V}}\deg\,(i)\}+r^-|\mathcal{V}|}{(1-r^-)(|\mathcal{V}|(|\mathcal{V}|-1)-|\mathcal{E}|)+r^-|\mathcal{V}|(|\mathcal{V}|-1)}\\&\leq 4\frac{(1-r^+)\{(|\mathcal{V}|-1)-\min_{i\in\mathcal{V}}\deg\,(i)\}+r^-(|\mathcal{V}|-1)}{(1-r^-)(|\mathcal{V}|(|\mathcal{V}|-1)-|\mathcal{E}|)+r^-|\mathcal{V}|(|\mathcal{V}|-1)}\\&=4\frac{1}{|\mathcal{V}|}\frac{(|\mathcal{V}|-1)-(1-r^-)\min_{i\in\mathcal{V}}\deg\,(i)}{(|\mathcal{V}|-1)-(1-r^-)\frac{|\mathcal{E}|}{|\mathcal{V}|}}.\end{aligned}\tag{97}$$

Here, since $\min_{i\in\mathcal{V}}\deg\,(i)\leq\frac{|\mathcal{E}|}{|\mathcal{V}|}$, the right hand side is decreasing with respect to $r$ and takes its maximum $4\frac{1}{|\mathcal{V}|}\frac{(|\mathcal{V}|-1)-\min_{i\in\mathcal{V}}\deg\,(i)}{(|\mathcal{V}|-1)-\frac{|\mathcal{E}|}{|\mathcal{V}|}}$ if $r=0$.

**(b)** (Skipgram type negative sampling case) Consider the setting in Example 1 (b). By the same discussion in (a), we have $\nu_1^+ = \frac{1}{2} \cdot \frac{(1-r)\max_{i\in\mathcal{V}}\deg(i)+r(|\mathcal{V}|-1)}{[(1-r)|\mathcal{E}|+r|\mathcal{V}|(|\mathcal{V}|-1)]}$ holds. Here, the right hand side is decreasing with respect to $r$ and takes its maximum $\frac{1}{2} \cdot \frac{\max_{i\in\mathcal{V}}\deg(i)}{|\mathcal{E}|}$ if $r = 0$.

The probability $\mathbb{P}\left[\left(i_{k,m}^-, j_{k,m}^-\right)\right]$ of $\left(i_{k,m}^-, j_{k,m}^-\right)$ appearing as the $k$-th negative edge is given as follows:

$$
\begin{aligned}
\mathbb{P}\left[\left(i_{k,m}^-, j_{k,m}^-\right)\right] &= \sum_{\left(i_{1,m}^+, j_{1,m}^+\right)\in\mathcal{V}\times\mathcal{V}} \mathbb{P}\left[\left(i_{k,m}^-, j_{k,m}^-\right)\middle|\left(i_{1,m}^+, j_{1,m}^+\right)\right] \\
&= \frac{\left[(1-r)\deg\left(i_{1,m}^-\right)+r(|\mathcal{V}|-1)\right]\left[(1-r)\deg\left(j_{1,m}^-\right)+r(|\mathcal{V}|-1)\right]}{\left[(1-r)|\mathcal{E}|+r|\mathcal{V}|(|\mathcal{V}|-1)\right]^2}.
\end{aligned}
\tag{98}
$$

Hence, the element in the $i$-th row and the $j$-th column of $\mathbb{E}_{\left(i_{k,m}^\bullet, j_{k,m}^\bullet\right)} \boldsymbol{E}_{i_{k,m}^\bullet, j_{k,m}^\bullet}^\top \boldsymbol{E}_{i_{k,m}^\bullet, j_{k,m}^\bullet}$ is given by

$$
\left[\mathbb{E}_{\left(i_{k,m}^-, j_{k,m}^-\right)} \boldsymbol{E}_{i_{k,m}^-, j_{k,m}^-}^\top \boldsymbol{E}_{i_{k,m}^-, j_{k,m}^-}\right]_{i,j} = \begin{cases} \frac{1}{2} \cdot \frac{(1-r^-)\deg(i)+r^-(|\mathcal{V}|-1)}{(1-r^-)|\mathcal{E}|+r^-|\mathcal{V}|(|\mathcal{V}|-1)} & \text{if } i = j, \\ 0 & \text{if } i \neq j. \end{cases}
\tag{99}
$$

for $\mathcal{Z} = \mathbb{L}^D, \mathbb{S}^D, \mathbb{I}^D$, and

$$
\begin{aligned}
&\left[\mathbb{E}_{\left(i_{k,m}^+, j_{k,m}^+\right)} \boldsymbol{E}_{i_{k,m}^+, j_{k,m}^+}^{\mathcal{Z}\ \top} \boldsymbol{E}_{i_{k,m}^+, j_{k,m}^+}^{\mathcal{Z}}\right]_{i,j} \\
&= \begin{cases} 2 \cdot \left[\frac{(1-r^-)\deg(i)+r^-(|\mathcal{V}|-1)}{(1-r^-)|\mathcal{E}|+r^-|\mathcal{V}|(|\mathcal{V}|-1)} - \frac{\left[(1-r)\deg\left(i_{1,m}^-\right)+r(|\mathcal{V}|-1)\right]\left[(1-r)\deg\left(j_{1,m}^-\right)+r(|\mathcal{V}|-1)\right]}{[(1-r)|\mathcal{E}|+r|\mathcal{V}|(|\mathcal{V}|-1)]^2}\right] & \text{if } i = j, \\ -2 \cdot \frac{\left[(1-r)\deg\left(i_{1,m}^-\right)+r(|\mathcal{V}|-1)\right]\left[(1-r)\deg\left(j_{1,m}^-\right)+r(|\mathcal{V}|-1)\right]}{[(1-r)|\mathcal{E}|+r|\mathcal{V}|(|\mathcal{V}|-1)]^2} & \text{if } i \neq j, \end{cases}
\end{aligned}
\tag{100}
$$

for $\mathcal{Z} = \mathbb{R}^D$. In other words, we have that

$$
\begin{aligned}
&\mathbb{E}_{\left(i_{k,m}^-, j_{k,m}^-\right)} \boldsymbol{E}_{i_{k,m}^-, j_{k,m}^-}^{\mathcal{Z}\ \top} \boldsymbol{E}_{i_{k,m}^-, j_{k,m}^-}^{\mathcal{Z}} \\
&= 2\left[\frac{(1-r^-)\operatorname{diag}(\boldsymbol{d}) + r^-(|\mathcal{V}|-1)\mathbf{I}}{(1-r^-)|\mathcal{E}|+r^-|\mathcal{V}|(|\mathcal{V}|-1)} - \frac{[(1-r^-)\boldsymbol{d}+r^-(|\mathcal{V}|-1)\mathbf{1}][(1-r^-)\boldsymbol{d}+r^-(|\mathcal{V}|-1)\mathbf{1}]^\top}{\left[(1-r^-)|\mathcal{E}|+r^-|\mathcal{V}|(|\mathcal{V}|-1)\right]^2}\right].
\end{aligned}
\tag{101}
$$

Hence, for $\mathcal{Z} = \mathbb{L}^D, \mathbb{S}^D, \mathbb{I}^D$, $\nu_k^- = \frac{1}{2} \cdot \frac{(1-r)\max_{i\in\mathcal{V}}\deg(i)+r(|\mathcal{V}|-1)}{[(1-r)|\mathcal{E}|+r|\mathcal{V}|(|\mathcal{V}|-1)]}$ holds for $k = 1, 2, \ldots, K^-$. Again, the right hand side is decreasing with respect to $r$ and takes its maximum $\frac{1}{2} \cdot \frac{\max_{i\in\mathcal{V}}\deg(i)}{|\mathcal{E}|}$ if $r = 0$. For $\mathcal{Z} = \mathbb{R}^D$, we have that

$$
\begin{aligned}
\nu_{k,\mathcal{Z}}^- &= \left\|2\left[\frac{(1-r^-)\operatorname{diag}(\boldsymbol{d}) + r^-(|\mathcal{V}|-1)\mathbf{I}}{(1-r^-)|\mathcal{E}|+r^-|\mathcal{V}|(|\mathcal{V}|-1)} - \frac{[(1-r^-)\boldsymbol{d}+r^-(|\mathcal{V}|-1)\mathbf{1}][(1-r^-)\boldsymbol{d}+r^-(|\mathcal{V}|-1)\mathbf{1}]^\top}{\left[(1-r^-)|\mathcal{E}|+r^-|\mathcal{V}|(|\mathcal{V}|-1)\right]^2}\right]\right\|_{\text{op},2} \\
&\leq 2\left[\frac{(1-r^-)\|\operatorname{diag}(\boldsymbol{d})\|_{\text{op},2} + r^-(|\mathcal{V}|-1)\|\mathbf{I}\|_{\text{op},2}}{(1-r^-)|\mathcal{E}|+r^-|\mathcal{V}|(|\mathcal{V}|-1)}\right. \\
&\qquad \left. + \frac{\left\|[(1-r^-)\boldsymbol{d}+r^-(|\mathcal{V}|-1)\mathbf{1}][(1-r^-)\boldsymbol{d}+r^-(|\mathcal{V}|-1)\mathbf{1}]^\top\right\|_{\text{op},2}}{\left[(1-r^-)|\mathcal{E}|+r^-|\mathcal{V}|(|\mathcal{V}|-1)\right]^2}\right] \\
&= 2\left[\frac{(1-r^-)\max_{i\in\mathcal{V}}\deg(i) + r^-(|\mathcal{V}|-1)}{(1-r^-)|\mathcal{E}|+r^-|\mathcal{V}|(|\mathcal{V}|-1)} + \frac{\|(1-r^-)\boldsymbol{d}+r^-(|\mathcal{V}|-1)\mathbf{1}\|_2^2}{\left[(1-r^-)|\mathcal{E}|+r^-|\mathcal{V}|(|\mathcal{V}|-1)\right]^2}\right] \\
&\leq 2\left[\frac{(1-r^-)\max_{i\in\mathcal{V}}\deg(i) + r^-(|\mathcal{V}|-1)}{(1-r^-)|\mathcal{E}|+r^-|\mathcal{V}|(|\mathcal{V}|-1)} + \frac{\|(1-r^-)\boldsymbol{d}+r^-(|\mathcal{V}|-1)\mathbf{1}\|_1^2}{\left[(1-r^-)|\mathcal{E}|+r^-|\mathcal{V}|(|\mathcal{V}|-1)\right]^2}\right] \\
&= 2\left[\frac{(1-r^-)\max_{i\in\mathcal{V}}\deg(i) + r^-(|\mathcal{V}|-1)}{(1-r^-)|\mathcal{E}|+r^-|\mathcal{V}|(|\mathcal{V}|-1)} + 1\right],
\end{aligned}
\tag{102}
$$

for $\mathcal{Z} = \mathbb{R}^D$, where the right hand side is decreasing with respect to $r^-$ and takes its maximum $2\left[\frac{\max_{i \in \mathcal{V}} \deg(i)}{|\mathcal{E}|} + 1\right]$ if $r^- = 0$.

$\square$

## D   Proof of Lemma 1

We recall Lemma 1.

**Lemma** (Lemma 1). *Suppose that $(\mathcal{V}, \mathcal{E})$ is a tree and $\delta^* : \mathcal{V} \times \mathcal{V} \to \mathbb{R}_{\geq 0}$ is given by its graph distance. Then, there exist $R \in \mathbb{R}_{\geq 0}$, representations $(z_1, z_2, \ldots, z_V) \in \bar{\mathcal{B}}_R$ in $\mathbb{L}^2$, and threshold $\theta_{\mathcal{Z}} \in \mathbb{R}$ that satisfy* (24) *for all $i, j \in \mathcal{V}$. In particular, $\mathcal{R}^{\mathcal{Z}}\left((z_v^*)_{v=1}^{|\mathcal{V}|}\right) = \mathcal{R}^*$.*

*Proof of Lemma 1.* Following [29], for a weighted graph $(\mathcal{V}, \mathcal{E})$ with a nonnegative weight function $w : \mathcal{E} \to \mathbb{R}_{\geq 0}$, take $\beta < \frac{\pi}{\max_v \deg(v)}$ and let $\mathfrak{n} := -2k \ln\left(\tan \frac{\beta}{2}\right)$ and $\eta := \max_{(i,j) \in \mathcal{E}} \frac{-2k \ln \tan \frac{\pi}{\max_{v=i,j} \deg(v)}}{w(i,j)}$. Here, $k$ is the absolute value of the curvature of the hyperbolic space that we consider. In this paper, $k = 1$ always holds. For $\epsilon > 0$, take $\tau$ so that it satisfies $\tau > \eta$ and $\tau \min_{(i,j) \in \mathcal{E}} w(i,j) > \mathfrak{n} \frac{1+\epsilon}{\epsilon}$. Then by the $(1 + \epsilon)$-distortion algorithm, we can obtain representations $z_1, z_2, \ldots, z_V \in \mathbb{L}^2$ such that

$$(1 - \epsilon)\tau \delta^*(i,j) < \Delta_{\mathbb{L}^2}(z_i, z_j) \leq (1 + \epsilon)\tau \delta^*(i,j), \tag{103}$$

for any $i, j \in \mathcal{V}$, where $\delta^*$ is the distance function on the weighted graph. Since $k = 1$ and we consider an unweighted graph, where $w(i,j) = 1$ holds for all $(i,j)$, $\mathfrak{n} := -2k \ln\left(\tan \frac{\beta}{2}\right)$, we have that $\mathfrak{n} := -2 \ln\left(\tan \frac{\beta}{2}\right)$ and $\eta < \mathfrak{n}$. Therefore, for $\tau > \mathfrak{n}\frac{1+\epsilon}{\epsilon}$, there exist representations that satisfy (103).

Let $\tau > \max\left\{\mathfrak{n}\frac{1+\epsilon}{\epsilon}, 2\right\}$, and $\theta_{\mathbb{L}^2} := 2\tau^2$ and by the $(1+\epsilon)$-distortion algorithm, obtain representations $z_1, z_2, \ldots, z_V \in \mathbb{L}^2$ such that they satisfy (103) for any $i, j \in \mathcal{V}$. Let $\epsilon := \frac{1}{4}$. Then, for $(i,j) \in \mathcal{E}$, we have that

$$\begin{aligned}
&\theta_{\mathbb{L}^2} - [\Delta_{\mathbb{L}^2}(z_i, z_j)]^2 \\
&\geq \theta_{\mathbb{L}^2} - (1+\epsilon)^2 \delta^*(i,j) \\
&\geq \tau^2\left(2 - \frac{25}{16}\right) \\
&\geq \frac{7}{4} \\
&> 1.
\end{aligned} \tag{104}$$

For $(i,j) \notin \mathcal{E}$, since $\delta^*(i,j) \leq 2$, we have that

$$\begin{aligned}
&[\Delta_{\mathbb{L}^2}(z_i, z_j)]^2 - \theta_{\mathbb{L}^2} \\
&\geq \tau^2(1-\epsilon)^2[\delta^*(i,j)]^2 - \theta_{\mathbb{L}^2} \\
&\geq \tau^2\left(\frac{9}{4} - 2\right) \\
&\geq 1.
\end{aligned} \tag{105}$$

Hence, the representations always satisfy the condition (24). If the diameter of the graph is $R_{(\mathcal{V}, \mathcal{E})}$, then we can get the representations in $\mathcal{B}_{\tau R_{(\mathcal{V}, \mathcal{E})}}$, which completes the proof. $\square$

## E   Proof of Lemma 2

We recall Lemma 2 below.

**Lemma** (Lemma 2). *Let $(\mathcal{V}, \mathcal{E})$ be a graph and define $W$ and $\rho$ by $W := |\mathcal{V}|(|\mathcal{V}| - 1)$ and $\mathcal{E} := \frac{|\mathcal{E}|}{W}$. Define $\mu := \min\{\mu^+, \mu^-\}$, where $\mu^+ := \frac{1}{W}\frac{1}{(1-r^+)\rho+r^+} - \frac{1}{W}\frac{r^-}{(1-r^-)(1-\rho)+r^-}$ and $\mu^- := \frac{1}{W}\frac{1}{(1-r^-)(1-\rho)+r^-} - \frac{1}{W}\frac{r^+}{(1-r^+)\rho+r^+}$. In LGE, the expected risk of the expected risk minimizers satisfies $\mathcal{R}^{\mathcal{Z}}\left((\boldsymbol{z}_v^*)_{v=1}^{|\mathcal{V}|}\right) \geq \mathcal{R}^* + V_{\min}^{\mathbb{R}^2}\mu$. Here, $V_{\min}^{\mathbb{R}^2}$ is not smaller than the number of disjoint 6-star subgraphs in the graph for the 2-dimensional LGE.*

*Proof of Lemma 2.* A truly positive pair appears as a positive pair in probability $\frac{1}{W}\frac{1}{(1-r^+)\rho+r^+}$ and as a negative pair in probability $\frac{1}{W}\frac{r^-}{(1-r^-)(1-\rho)+r^-}$. Also, a truly negative pair appears as a negative pair in probability $\frac{1}{W}\frac{1}{(1-r^-)(1-\rho)+r^-}$ and as a positive pair in probability $\frac{1}{W}\frac{r^+}{(1-r^+)\rho+r^+}$.

Hence, each pair that violates the condition (24) increases the expected risk by $\mu := \min\{\mu^+, \mu^-\}$, where $\mu^+ := \frac{1}{W}\frac{1}{(1-r^+)\rho+r^+} - \frac{1}{W}\frac{r^-}{(1-r^-)(1-\rho)+r^-}$ and $\mu^- := \frac{1}{W}\frac{1}{(1-r^-)(1-\rho)+r^-} - \frac{1}{W}\frac{r^+}{(1-r^+)\rho+r^+}$. Hence, we have that $\mathcal{R}^{\mathcal{Z}}\left((\boldsymbol{z}_v^*)_{v=1}^{|\mathcal{V}|}\right) \geq \mathcal{R}^* + V_{\min}^{\mathbb{R}^2}\mu$. Also, the achievable minimum expected risk $\mathcal{R}^*$ is $\rho\frac{r^-}{(1-r^-)(1-\rho)+r^-}$ and $(1-\rho)\frac{r^+}{(1-r^+)\rho+r^+}$, which is attained if the condition (24) is always satisfied.

In the following, we prove that $V_{\min}^{\mathbb{R}^2}$ is no smaller than the number of 6-star subgraphs. The following proves that for each 6-star subgraph, there is a pair that violates (5). Let $c$ be the center of the 6-star subgraph and $v_1, v_2, \ldots, v_6$ be its neighborhood. Define $\delta_m^* := \delta^*(c, v_m)$. Let $B_\delta(\boldsymbol{z})$ be an open ball of radius $\delta$ centered at $\boldsymbol{z}$ in $\mathbb{R}^2$. Assume that $\boldsymbol{z}_c, \boldsymbol{z}_{v_1}, \ldots, \boldsymbol{z}_{v_6} \in \mathbb{R}^2$ satisfies (5) and define $\Delta_m := \Delta_{\mathbb{R}^2}(\boldsymbol{z}_c, \boldsymbol{z}_{v_m})$. For $m, m' = 1, 2, \ldots, 6$, $\boldsymbol{z}_{v_m} \in B_{\Delta_{m'}}(\boldsymbol{z}_c)$ and $\boldsymbol{z}_{v_m} \notin B_{\Delta_{m'}}(\boldsymbol{z}_{v_{m'}})$ are necessary. Hence, $\angle \boldsymbol{z}_{v_m}\boldsymbol{z}_c\boldsymbol{z}_{v_{m'}} > 60°$. Thus, segments $\boldsymbol{z}_c\boldsymbol{z}_{v_1}, \boldsymbol{z}_c\boldsymbol{z}_{v_2}, \ldots, \boldsymbol{z}_c\boldsymbol{z}_{v_6}$ partition $360°$ into 6 angles larger than $60°$, which is contradiction. $\qquad\square$

# F  Proof of Proposition 2

We recall the proposition below.

**Proposition** (Proposition 2). *Suppose that $(\mathcal{V}, \mathcal{E})$ is a tree and $\delta^* : \mathcal{V} \times \mathcal{V} \to \mathbb{R}_{\geq 0}$ is given by its graph distance, and take $R$ given in Lemma 1. Let $\nu_{1,\mathbb{L}^D} := \left(\sqrt{\nu_{1,\mathbb{L}^D}^+} + \sqrt{\nu_{1,\mathbb{L}^D}^-}\right)^2$. If $M > \left(\frac{3\omega_{\mathbb{L}^D}(R)}{4|\mathcal{V}|\mu V_{\min}^{\mathbb{R}^2}}\left(\sqrt{8\nu_{1,\mathbb{L}^D}\ln|\mathcal{V}|} + \sqrt{\ln\frac{2}{\delta}}\right) + \frac{1}{2\left(\sqrt{8\nu_{1,\mathbb{L}^D}\ln|\mathcal{V}|} + \sqrt{\ln\frac{2}{\delta}}\right)}\right)^2$, then HGE's $\mathcal{R}^{\mathcal{Z}}\left((\boldsymbol{z}_v^*)_{v=1}^{|\mathcal{V}|}\right)$ is smaller than LGE's.*

*Proof.* If $a, b, c > 0$ and assume $x > 0$, then $ax^2 + bx < c \Leftrightarrow 0 < x < -b + \sqrt{b^2 + 4ac}$. We are going to find $M$ that satisfies

$$\frac{2\omega_{\mathbb{L}^D}(R)}{M}\sum_{\bullet=+,-}\sum_{k=1}^{K^\bullet} L_k^\bullet\left(\sqrt{2M\nu_{k,\mathbb{L}^D}^\bullet \ln|\mathcal{V}|} + \frac{1}{6}\ln|\mathcal{V}|\right) + I_l(\mathcal{B})\sqrt{\frac{\ln\frac{2}{\delta}}{M}} < \mu V_{\min}^{\mathbb{R}^2}, \qquad (106)$$

where $K^+ = K^- = 1$, $L_1^+ = L_1^- = 1$, and $I_l = 1$. Since we consider $R$ that satisfies the condition in Lemma 2, $2\omega_{\mathbb{L}^D}(R) \leq I_l(\mathcal{B})$ is valid. Hence, we consider the following condition.

$$\omega_{\mathbb{L}^D}(R)\left(2\sqrt{2\frac{1}{M}\nu_{1,\mathbb{L}^D}\ln|\mathcal{V}|} + 4\cdot\frac{1}{6}\frac{1}{M}\ln|\mathcal{V}| + \sqrt{\frac{\ln\frac{2}{\delta}}{M}}\right) < \mu V_{\min}^{\mathbb{R}^2}. \qquad (107)$$

This corresponds to $x = \frac{1}{\sqrt{M}}$, $a = \frac{2}{3}|\mathcal{V}|$, $b = \sqrt{8\nu_{1,\mathbb{L}^D}\ln|\mathcal{V}|} + \sqrt{\ln\frac{2}{\delta}}$, $c = \frac{\mu V_{\min}^{\mathbb{R}^2}}{\omega_{\mathbb{L}^D}(R)}$. We can formulate the condition as follows:

$$0 < \frac{1}{\sqrt{M}} < -b + \sqrt{b^2 + 4ac}$$
$$\Leftrightarrow M > \frac{1}{\left(-b + \sqrt{b^2 + 4ac}\right)^2}. \qquad (108)$$

Here, we have that

$$\frac{1}{\left(-b+\sqrt{b^2+4ac}\right)^2} = \frac{\left(b+\sqrt{b^2+4ac}\right)^2}{(4ac)^2}$$

$$\leq \frac{\left(b+b\left(1+\frac{2ac}{b^2}\right)\right)^2}{(4ac)^2}$$

$$= \left(\frac{b\left(2+\frac{2ac}{b^2}\right)}{4ac}\right)^2$$

$$= \left(\frac{b}{2ac}+\frac{1}{2b}\right)^2.$$

$$\leq \left(\frac{b}{2ac}+\frac{1}{2b}\right)^2.$$

$$= \left(\frac{3\omega_{\mathbb{L}^D}(R)}{4|\mathcal{V}|\mu V_{\min}^{\mathbb{R}^2}}\left(\sqrt{8\nu_{1,\mathbb{L}^D}\ln|\mathcal{V}|}+\sqrt{\ln\frac{2}{\mathfrak{d}}}\right)+\frac{1}{2\left(\sqrt{8\nu_{1,\mathbb{L}^D}\ln|\mathcal{V}|}+\sqrt{\ln\frac{2}{\mathfrak{d}}}\right)}\right)^2,$$

$$(109)$$

which completes the proof. Here, the first inequality holds since $\sqrt{1+y}\leq 1+\frac{1}{2}y$. $\qquad\square$

## G   Proof of Example 4

*Proof.* Example 4 We obtain the result by applying Proposition 2. Here, $|\mathcal{V}|=\frac{1-\lambda^h}{1-\lambda}$, $|\mathcal{E}|=|\mathcal{V}|-1$. The radius $R$ is given by the proof of Lemma 1, $\nu_{1,\mathbb{L}^D} := \left(\sqrt{\nu_{1,\mathbb{L}^D}^+}+\sqrt{\nu_{1,\mathbb{L}^D}^-}\right)^2$ is given by Proposition 1, and $\mu$ is given by the first half of Lemma 2. From the second half of Lemma 2, we have that $V_{\min}^{\mathbb{R}^2} > \lambda^2$ if $\lambda \geq 5$ and $h \geq 4$. $\qquad\square$