# OpenReview forum: "Generalization Bounds for Graph Embedding Using Negative Sampling: Linear vs Hyperbolic"
_NeurIPS.cc/2021/Conference — NeurIPS 2021 Poster_

### Official Review · Reviewer_ywBH · 2021-07-06

**Rating:** 7
**Confidence:** 3

**Summary:**

The paper presents a generalization error bound for graph embeddings in the defined dissimilarity space, including the linear space and the hyperbolic space. The main conclusion is that error is polynomial and exponential with respect to the embedding space's radius in linear and hyperbolic spaces respectively. However, hyperbolic graph embeddings (HGE) with sufficiently larger number of data represents hierarchical data more effectively than linear graph embeddings (LGE).

**Limitations And Societal Impact:**

Yes, the authors addressed the limitations of the work. The potential negative societal impact is not addressed, but I think there is no such concerns of this work.

**Main Review:**

The paper has solid theoretical contributions. The assumptions and preliminaries are presented very clearly and detailedly. The derivation of the generalization error bound is also clear. The discussion on the bias-variance trade-off of HGE and LGE is interesting and insightful.

Some questions:
1. The generalization bounds rely on the assumptions on embedding spaces's radius. However, the hyperbolic embeddings of the hyperboloid model (which is the hyperbolic model analyzed in the paper) may have very large norms in practical experiments, especially when the input graph is a very large tree, then the leaves of the tree will be embedded into the points much far away from the origin. The assumptions on the embedding space's radius do not seem practical in such case.
2. In Remark 5, why is the bound in Theorem 1 for $\mathcal{Z}=\mathbb{R}^D$ or $\mathbb{I}^D$ includes a term $R^2L_k^\cdot$? The term in Eq. (15) is $\omega_{\mathcal{Z}}(R)L_k^\cdot$, which should be $RL_k^\cdot$ since $\omega_{\mathbb{R}^D}=\omega_\mathbb{I}^D=(2R)$?

Some typos:
1. Eq. (4), $(i\cdot v_1),(v_1\cdot v_2),\dots,(v_{K-1}\cdot j))$ --> $(i,v_1),(v_1,v_2),\dots,(v_{K-1},j)$.
2. Line 154, are consists of --> consist of.
3. Line 165, $s_m^+$ and $s_m^+$ --> $s_m^+$ and $s_m^-$.
4. Line 308, the balanced --> the more balanced.
5. Eq. (24), the second $(i,j)\in\mathcal{E}$ --> $(i,j)\notin\mathcal{E}$.
6. Line 327, $(i_{k,m}^-,j_{k,m}^-)\in\mathcal{E}$ --> $(i_{k,m}^-,j_{k,m}^-)\notin\mathcal{E}$.

**Time Spent Reviewing:**

5

---

> ### Author Response · Authors · 2021-08-11
> **Author response**
>
> ### **Comment 1**
> > The generalization bounds rely on the assumptions on embedding spaces's radius. However, the hyperbolic embeddings of the hyperboloid model (which is the hyperbolic model analyzed in the paper) may have very large norms in practical experiments, especially when the input graph is a very large tree, then the leaves of the tree will be embedded into the points much far away from the origin. The assumptions on the embedding space's radius do not seem practical in such case.
> $\newcommand{\bbI}{\mathbb{I}}$
> $\newcommand{\bbR}{\mathbb{R}}$
>
> **Response:** Representations in the hyperboloid model have very large "norms" because the model has property of making the "norm" exponential with respect to the distance from the origin. However, this property has no relation to the assumption of Theorem 1. This is because the embedding space's radius $R$ in (14) is measured by the distance function, not by the "norm" defined by the coordinate system of a particular model such as the hyperboloid model. Hence, our theory is independent of which hyperbolic space model to choose, and does not suffer from very large "norm" problem caused by the hyperboloid model's property.
>
> To further understand that the hyperboloid model's property does not influence the radius $R$, we also discuss the concrete form of the distance function as follows. The exponentially large "norm" caused by the hyperboloid model is cancelled out in evaluating the radius as we measure the radius by the distance function $\Delta_{\mathbb{L}^{D}} (\boldsymbol{x}, \boldsymbol{x}') := \mathop{\mathrm{arcosh}} (- \langle \boldsymbol{x}, \boldsymbol{x}' \rangle_\mathrm{M})$. This is because the function is almost logarithmic with respect to its operands, according to the property of $\mathrm{arcosh}$.
> Thus, the hyperboloid model's property of making the "norm" exponential does not influence $R$.
>
> ### **Comment 2**
> > In Remark 5, why is the bound in Theorem 1 for $\mathcal{Z} = \mathbb{R}^{D}$ or $\mathbb{I}^{D}$ includes a term $R^{2} L_{k}$? The term in Eq. (15) is
> $\omega_{\mathcal{Z}} (R) L_{k}$, which should be $R L_{k}$ since $\omega_{\mathcal{\bbR^{D}}} (R) = \omega_{\mathcal{\bbI^{D}}} (R) = (2R)$?
>
> **Response:** Equations $\omega_{\mathcal{\bbR^{D}}} (R) = \omega_{\mathcal{\bbI^{D}}} (R) = (2R)$, which are the definitions of $\omega_{\mathcal{\bbR^{D}}}$ and $\omega_{\mathcal{\bbI^{D}}}$, included a typo indeed and sorry for the typo. The definitions should be $\omega_{\mathcal{\bbR^{D}}} (R) = \omega_{\mathcal{\bbI^{D}}} (R) = (2R)^2$, where index 2 was missing in the original paper.
> Accordingly, the bound includes a term $R^{2} L_{k}$ for $\mathcal{Z} = \bbR^{D}$ or $\bbI^{D}$, which is consistent to Remark 5. We will correct the typo ($(2R)$ -> $(2R)^2$) as well as those that you have pointed out in the final version.

---

> > ### Comment · Reviewer_ywBH · 2021-08-17
> > **The author response addressed my comments very well**
> >
> > The author response addressed my comments very well. I also found the authors addressed many concerns of my fellow reviewers clearly. I would like to keep my score (7) since I think this is a good paper and should be accepted.

---

> > > ### Author Response · Authors · 2021-09-03
> > > **Thank you for the reply.**
> > >
> > > Thank you for the reply emphasizing the goodness of our work.

---

### Official Review · Reviewer_WFTn · 2021-07-15

**Rating:** 5
**Confidence:** 5

**Summary:**

Graph embedding has already been a common and effective technique in many applications, while the corresponding theoretical analysis is limited to ideal noiseless settings. This work intends to derive the unified generalization error bound for negative-sampling-based graph embedding regarding the existence of noise. However, the assumption made in this work to drive the theoretical analysis is somewhat impractical making the derived bound a little bit vacuous.

**Limitations And Societal Impact:**


Yes

**Main Review:**

Strength
The motivation of this paper is reasonable. This paper derives the generalization error bound for negative-sampling-based graph embedding which is applicable to various settings (say different embedding spaces, data distributions, and loss functions).

Weakness：

Nonetheless, some technical details are based on very strong assumptions and the risk defined in the paper cannot reflect the true embedding ability of a model. This too suggests that bounding the excess risk therein cannot guarantee a good generalization performance.


1)	The assumption that all positive-negative pair sequences are i.i.d is too strong, which is not applicable for practical applications. This is because each sequence has $K^+$ positive pairs and $K^-$ negative pairs, and thus it is difficult to ensure that these pairs have no interactions and are independent of each other. Therefore, the generalization error bound is limited and may not make sense.

2)	I’m wondering that if tighten result could be derived if the telescoping-based Lemma 3 is replaced by the Vector-Contraction Inequality proposed in the following paper:
A Vector-Contraction Inequality for Rademacher Complexities. ALT 2016: 3-17

3)	The authors should re-organize the proof to make it easy to follow.

4)	There are many inconsistent/unclaimed symbols making the unnecessary confusion. For example,
a)	In line 190, $i_{1,m}^-$ should be $i_{1,m}^+$;
b)	In line 195, $r$ is not claimed;
c)	In Eq.(31), $R_M(F)$ is not claimed;
d)	In line 579, $D$ should be $D_k$, etc.
In addition, the symbols should be as concise as possible to make the paper clear and easy to understand.’

5)	The current version of this paper is not easy to understand, and the writing and structure need to be further improved. For example,
a)	In definition 1, (b) should be in another line;
b)	In line 543, “By ... the its…”, ‘the’ should be removed;
c)	In line 569, “Let D is …” should be “Let D be…”;
d)	In theorem 3, reference [41] is not included in the main paper.
e)	Many references have been published in conference such as [21] instead of arXiv.

**Time Spent Reviewing:**

6h

---

> ### Author Response · Authors · 2021-08-10
> **Author response**
>
> $\newcommand{\nc}[3][0]{\newcommand{#2}[#1]{#3}}
> \nc{\bbZ}{\mathbb{Z}}\nc{\bbR}{\mathbb{R}}\nc{\bbE}{\mathbb{E}}\nc{\Zg}{\bbZ_{\ge 0}}
> \nc{\calA}{\mathcal{A}}\nc{\calG}{\mathcal{G}}\nc{\calV}{\mathcal{V}}
> \nc{\calE}{\mathcal{E}}\nc{\calX}{\mathcal{X}}\nc{\calY}{\mathcal{Y}}
> \nc{\calF}{\mathcal{F}}\nc{\calD}{\mathcal{D}}
> \nc{\frakR}{\mathfrak{R}}\nc{\scrG}{\mathscr{G}}
> \nc{\tilf}{\tilde{f}}
> \nc{\xprm}{x'}\nc{\yprm}{y'}
> \nc{\bsf}{\boldsymbol{f}}\nc{\bsy}{\boldsymbol{y}}\nc{\bsc}{\boldsymbol{c}}
> \nc{\cAp}{\calA^{+}}\nc{\cAn}{\calA^{-}}
> \nc{\xmparen}{(x_m)}\nc{\xmseq}{\xmparen_{m=1}^{M}}
> \nc{\sigmparen}{(\sigma_m)}\nc{\sigmseq}{\sigmparen_{m=1}^{M}}
> \nc{\sigkmparen}{(\sigma_{k, m})}\nc{\sigkmseq}{\sigkmparen_{k=1, m=1}^{K, M}}
> \nc[1]{\norm}{\lVert#1\rVert_{2}}
> \nc[3]{\list}{#1,#2,\dots,#3}
> \nc[2]{\seq}{#1_{1},#1_{2},\dots,#1_{#2}}
> \nc[2]{\pr}{(i_{#1}^{#2}, j_{#1}^{#2})}
> \nc[2]{\prs}{\left(\pr{1}{#2},\pr{2}{#2},\dots,\pr{#1}{#2}\right)}
> \nc[2]{\prsm}{\pr{1,m}{#2},\pr{2,m}{#2},\dots,\pr{#1,m}{#2}}
> \nc{\Kp}{{K^+}}\nc{\Kn}{{K^-}}\nc{\kp}{{\kappa^+}}\nc{\kn}{{\kappa^-}}
> \nc{\qq}{\qquad}
> \nc[1]{\tbf}{\textbf{#1}}\nc{\ntv}{\tbf{node2vec}}\nc{\ntvw}{\tbf{node2vecWalk}}
> \nc{\Exsm}{\bbE_{\xmseq, \sigmseq}}\nc{\Exskm}{\bbE_{\xmseq, \sigkmseq}}
> \nc{\sumk}{\sum_{k=1}^{K}}\nc{\summ}{\sum_{m=1}^{M}}
> \nc{\supf}{\sup_{\bsf \in \calF}}\nc{\supfk}{\sup_{f_k \in \calF_k}}
> \nc{\tfx}{\tilf_k (x_{k,m})}
> \nc[2]{\itr}{\tbf{For } #1 \gets 1,2,\dots,#2:}
> \nc{\VV}{(\calV \times \calV)}
> $
> **Response to Weakness 1:** Our main assumption (Assumption 1) is actually applicable to many practical applications using negative sampling, including DeepWalk [6], LINE [7], node2vec [9], Poincare Embedding [20], as we show in the next section (Applications).
> The main reason of its wide applicability is that we do **not** assume that **pairs** $\prsm{\Kp}{+}, \prsm{\Kn}{-}$ in $s_m$ are i.i.d., while we only assume that **pair sequences** $s_1, s_2, \dots, s_M$ are i.i.d., as we emphasized in Remark 2.
> For example, the assumption applicable even if the distributions of $\pr{1,m}{+}, \pr{1,m}{-}, \pr{2,m}{-}$ depends on each other for each $m$.
> For this reason, even though each sequence $s_m$ has $\Kp$ positive pairs and $\Kn$ negative pairs, it is not difficult to ensure that $s_m$ and $s_{m'}$ for $m \ne m'$ have no interactions and are independent of each other for many applications.
> To wrap up, our assumption is not too strong in the sense that it covers many practical existing applications, but not too weak in the sense that it can guarantee the convergence of the error as we can see in Theorem 1.
>
> As mentioned above, we show our assumption's wide applicability to practical applications by giving specific examples such as [6] [7] [9] [20] in the next section.
> Note that we omitted this discussion from the limited space of the paper since it is not directly related to the paper's prior objective, the comparison between LGE and HGE.
>
> **Applications**
>
> We discuss Assumption 1's applicability based on Algorithm R3.1 below, since it generalizes data sampling algorithms in [6] [7] [9] [20] and clarifies the sufficient condition for Assumption 1 to be satisfied. First, we introduce necessary definitions, then provide Algorithm R3.1's pseudocode.
> - **Definitions**
>   - $K \Zg$: the set of nonnegative multiples of $K \in \Zg$
>   - $s + s' := (\seq{x}{K}, \seq{\xprm}{K'})$, where $\calX$ is set, $K, K' \in \Zg$, $s = (\seq{x}{K}) \in \calX^K$ and $s' = (\seq{\xprm}{K'}) \in \calX^{K'}$.
>   - We say an algorithm is *stochastic* if it returns a random value whose distribution depends on the algorithm's inputs.
>   - We say a stochastic algorithm (SA) is *memoryless* if the distribution of its output is determined only by the algorithm's inputs.
>   - $\cAp$: a SA for positive sampling
>     - Inputs: $\calG$: a graph, $\Kp$: # positive pairs,
>     - Output: $\Kp$ positive pairs.
>   - $\cAn$: a SA for negative sampling
>     - Inputs: $\calG$: a graph, $\pr{}{+}$: a positive pair, $\kn$: # negative pairs.
>     - Output: $\kn$ negative pairs.
>
> ----
> **Algorithm R3.1**
>
> ----
> $\tbf{Input}$ $\calG=(\calV,\calE)$: graph, $M\in\Zg$: # iterations, $\cAp$: positive sampling algorithm, $\cAn$: negative sampling algorithm, $\Kp\in\Zg$: # positive pairs, $\Kn\in\Kp\Zg$: # negative pairs.
>
> $\kn \gets \frac\Kn\Kp$
>
> $s \gets ()$
>
> $\itr{m}{M}$
>
> $\qq \prs{\Kp}{+} \gets \cAp (\calG,\Kp)$
>
> $\qq s_m^+ \gets \prs{\Kp}{+}$
>
> $\qq s_m^- \gets ()$
>
> $\qq \itr{k}{\Kp}$
>
> $\qq \qq \prs{\kn}{-} \gets \cAn \left(\calG,\pr{k}{+},\kn,\right)$
>
> $\qq \qq s_m^- \gets s_m^- + \prs{\kn}{-}$
>
> $\qq s_m \gets (s_m^+, s_m^-)$
>
> $\qq s \gets s + (s_m)$
>
> $\tbf{Output } s \in \left(\VV^\Kp, \VV^\Kn \right)^M$: training data
>
> ----
> According to Algorithm R3.1, the outputs of $\cAp$ and $\cAn$ in the $m$-th outer loop only affect $s_m$.
> Hence, **if $\cAp$ and $\cAn$ are memoryless, then training data distribution by Algorithm R3.1 satisfies Assumption 1.**
>
> For this reason, it is sufficient to confirm that $\cAp$ and $\cAn$ are memoryless. Here, $\cAn$ is usually memoryless, because a trial of negative sampling to output $\kappa^-$ pairs is conducted mechanically and is independent of another trial.
> Thus, the remainder to discuss is whether $\cAp$ is memoryless or not. We discuss this in the following.
> - **Examples where positive pairs are sampled independently** such as in LINE [7] and Poincare embedding [20], where the dependency among positive pairs is not considered. These cases simply correspond to Algorithm R3.1 with memoryless $\cAp$ and $K^+ = 1$.
> - **Examples where positive pairs are sampled by a random walk (RW)** as in node2vec [9] and DeepWalk [6] with negative sampling. Sampling positive pairs by a RW causes dependency among them. Let us discuss node2vec as an example. node2vec's sampling algorithm is given as a special case of Algorithm R3.1 with $\cAp=\cAp_\ntv$, which is given as follows.
>
>   ----
>   **Algorithm** $\cAp_\ntv$
>
>   ----
>   $\tbf{Input } \calG,\cAp,\Kp\in\Zg$
>
>   $\kp \gets \frac\Kp{|\calV|}$
>
>   $s^+ \gets ()$
>
>   $\tbf{For all } v \in \calV$
>
>   $\qq \prs{\kp}{+} \gets \cAp_\ntvw (\calG,v,\kp)$
>
>   $\qq s^+ \gets s^+ + \prs{\Kp}{+}$
>
>   $\tbf{Output } s^+ \in \VV^\Kp$
>
>   ----
> Here, $\cAp_\ntvw$ is a RW-based SA, whose specific form is given as $\ntvw$ in [9], where $\calG, v, \kappa^{+}$ here corresponds to $G, v, l$ in the paper.
> Important is that although the positive pairs output by each call of $\cAp_\ntvw$ are dependent on each other, $\cAp_\ntvw$ is still memoryless because an independent RW runs for each call.
> Hence, the training data distribution of node2vec satisfies Assumption 1. A similar discussion holds for DeepWalk [6] with negative sampling.
>
> **Response to Weakness 2:** No. The bound obtained by the vector contraction inequality (VCI) is looser than ours, as proved below.
>
> Let us review the VCI. Let $\calX$ be a data domain, $\seq{x}{M}$ be $\calX$-valued i.i.d. random variables, $\sigmseq$ and $\sigkmseq$ be sequences of i.i.d. random variables that take a value -1 or +1 in the equal probability 0.5, $\calF \subset \{\bsf| \bsf: \calX \to \bbR \}$, and $\seq{l}{M}: \bbR^K \to \bbR$. Let $\norm{\bsy} := \sqrt{\bsy^\top \bsy}$. Let $L$ be the maximum in the Lipschitz constants of $\seq{l}{M}$, defined by
> $$
> L := \max_{m \in [M]} \sup_{\bsy, \bsy' \in \bbR^K} \frac{|l_m (\bsy) - l_m(\bsy')|}{\norm{\bsy - \bsy'}}.
> $$
> The VCI states that
> $$
> \Exsm\supf\summ\sigma_m l_m (\bsf (x_m)) \le \sqrt2 L \Exskm\supf\sumk\summ\sigma_{k,m} f_k (x_m).
> $$
>
> While the VCI holds under very general conditions, the paper discussed the following special case.
> - **Conditions R3.1**
>   - $\exists l: \bbR^K \to \bbR$ such that $l=l_1=l_2=\dots=l_M$.
>   - $\exists \seq{\calX}{K}$ such that $\calX=\calX_1 \times \calX_2 \times \dots \times \calX_K$.
>   - For all $k \in [K]$, $f_k (x_m)$ only depends on the $k$-th element $x_{k, m}$ of $x_{m}$, where $x_m=(x_{1,m},x_{2,m},\dots,x_{K,m})$, that is, there exists $\tilf_k: \calX_k \to \bbR$ such that $f_k (x_m) = \tfx$.
>
> First, let us consider what the VCI gives under these conditions. By the VCI, we have that
> $$
> \begin{split}
> \Exsm\supf\summ\sigma_m h (\bsf (x_m))
> & \le \sqrt2 L \Exskm\supf\sumk\summ\sigma_{k,m} \tfx\newline
> & \le \sqrt2 L \Exskm\supfk\sumk\summ\sigma_{k,m} \tfx\newline
> & \le \sqrt2 \sumk L \Exsm\supfk\summ\sigma_m \tfx,
> \end{split}
> \tag{R3.1}
> $$
> where $\calF_k$ is defined by (37) in the supplementary matrial (SM).
>
> Let's compare this result with our result by Lemma 3 in SM. Lemma 3 states that
> $$
> \Exsm\supf\summ\sigma_m h (\bsf (x_m))
> \le \sumk L_k \Exsm\supfk\summ\sigma_m f_k (x_{k,m}).
> \tag{R3.2}
> $$
> Here $L_{k}$ is the Lipschitz constant of $h$ with respect to the $k$-th element given by
> $$
> L_k = \sup_{\bsy \in \bbR^K, \yprm_k \in \bbR} \frac{|h(\begin{bmatrix} y_1 & y_2 & \cdots & y_k & \cdots & y_K \end{bmatrix}^\top) - h(\begin{bmatrix} y_1 & y_2 & \cdots & \yprm_k & \cdots & y_K \end{bmatrix}^\top)|}{|y_k - \yprm_k|}.
> $$
> Since $L \ge L_k$ always holds, our bound is not worse than that by the VCI.
> In particular, if $h(\bsy) = \bsc^\top \bsy$, where each element of $\bsc$ is either -1 or +1, then $L=\sqrt{K}$ and $L_k = 1$. Hence, the bound by the VCI is $O(\sqrt{K})$, while ours is $O(1)$, with respect to $K$. The dependency on other variables are the same. A similar discussion holds for the loss function given by Example 2 (a). This is not surprising because the VCI is for more general cases, and our lemma explicitly uses Conditions R3.1.
>
> **Response to Weakness 3,4,5:**
> Following your suggestions (from 3 to 5-d) to enhance the clarity, such as reorganizing proofs and correcting typos (e.g., $r$ -> $r^+$ in line 195, $\frakR_M$ -> $\frakR_{\calD,M}$ in (31)), we will make modification accordingly in the final version. For 5-e, while we cited two arXiv papers [21] [35], we have double-checked that our citation was correct since [35] has not been published yet and [21] was published in ICML after this NeurIPS's deadline.

---

> > ### Comment · Reviewer_WFTn · 2021-08-25
> > **Reply to the Authors**
> >
> > I would like to thank the authors for their detailed reply, which addresses some of my concerns. But I still believe that the readability should be improved.  So I will raise my score to 5.

---

> > > ### Author Response · Authors · 2021-09-03
> > > **Thank you for the reply**
> > >
> > > We appreciate your comments confirming that our reply has addressed your concerns. We will further improve the readability of the final version by reflecting on your detailed and specific feedback.

---

### Official Review · Reviewer_WZeR · 2021-07-17

**Rating:** 7
**Confidence:** 3

**Summary:**

This paper presents a generalization error bound for graph embedding with negative sampling. The theoretical results are general and can be applied to various settings, e.g., the embedding space, data distribution, and loss function. The theoretical results show the radius of embedding space plays a key role in graph embedding error, and show the imbalanced data distribution can worsen the error. This paper provides specific error bounds for many negative sampling strategies.

**Limitations And Societal Impact:**

The authors adequately addressed the limitations and potential negative societal impact of their work.

**Main Review:**

This paper studies a very interesting problem that is overlooked in many works. The proposed error bound for negative sampling-baed graph embedding is practical in real-world applications. The theoretical results are impressive and can benefit researchers to choose embedding space for one specific graph embedding task.

It's not clear why the theorem is applicable for the inner product space. the dissimilarity function of inner product space is different from the one in Euclidean space, so that loss function and the expected risk are also different.


**Time Spent Reviewing:**

2 hours

---

> ### Author Response · Authors · 2021-08-10
> **Author response**
>
> ### **Comment**
> > It's not clear why the theorem is applicable for the inner product space. the dissimilarity function of inner product space is different from the one in Euclidean space, so that loss function and the expected risk are also different.
> $\newcommand{\bbE}{\mathbb{E}}$
> $\newcommand{\bbR}{\mathbb{R}}$
> $\newcommand{\bbI}{\mathbb{I}}$
> $\newcommand{\bbS}{\mathbb{S}}$
> $\newcommand{\bbL}{\mathbb{L}}$
> $\newcommand{\bbRD}{\bbR^{D}}$
> $\newcommand{\bbID}{\bbI^{D}}$
> $\newcommand{\bbSD}{\bbS^{D}}$
> $\newcommand{\bbLD}{\bbL^{D}}$
> $\newcommand{\bsE}{\boldsymbol{E}}$
> $\newcommand{\bsG}{\boldsymbol{G}}$
> $\newcommand{\zvparen}{(z_{v})}$
> $\newcommand{\zvseq}{\zvparen_{v=1}^{V}}$
> $\newcommand{\calZ}{\mathcal{Z}}$
> $\newcommand{\rmdiag}{\mathrm{diag}}$
> $\newcommand{\rmoff}{\mathrm{off}}$
>
> **Response:** Although Theorem 1 consists of a single inequality (15), we obtain **different** upper bounds for Euclidean space $\bbRD$ and inner product space $\bbID$ (and $\bbSD$ and $\bbLD$) by substituting $\calZ$ with $\bbRD$ and $\bbID$.
> The difference originates from $a_\rmdiag^{\calZ}$ and $a_\rmoff^{\calZ}$.
> Specifically, $(a_\rmdiag^{\bbRD}, a_\rmoff^{\bbRD}) = (+2, -2)$, while $(a_\rmdiag^{\bbID}, a_\rmoff^{\bbID}) = (+ \frac{1}{2}, 0)$ as in line 271.
> Hence, $\nu_{k \calZ}^{\bullet} := \bbE_{(i_{k, m}^{\bullet}, j_{k, m}^{\bullet})} {\bsE_{(i_{k, m}^{\bullet}, j_{k, m}^{\bullet})}^{\calZ}}^\top {\bsE_{(i_{k, m}^{\bullet}, j_{k, m}^{\bullet})}^{\calZ}}$ is different for $\calZ = \bbRD$ and $\bbID$, since $\bsE_{(i_{k, m}^{\bullet}, j_{k, m}^{\bullet})}^{\calZ}$ depends on $a_\rmdiag^{\calZ}$ and $a_\rmoff^{\calZ}$, as we can see in (16).
> As a result, the right hand side of (15) is also different for $\calZ = \bbRD$ and $\bbID$. Thus, we get different upper bounds for $\bbRD$ and $\bbID$.

---

### Official Review · Reviewer_ixpa · 2021-07-18

**Rating:** 7
**Confidence:** 3

**Summary:**

This paper studies the question of the generalization error incurred when embedding graphs in linear vs. hyperbolic space. Recent literature (e.g. work of Nickel and Kiela, ref. [20]) has shown that graph data with hierarchical structure is often better embedded in hyperbolic space--due to its higher representation capacity--than in linear space (for which it becomes necessary to increase the dimensionality of the embedding). However, this choice may come with a tradeoff in terms of generalization error, and the current paper studies this phenomenon in the context of embeddings obtained via negative sampling. Specifically, the authors show that the generalization errors of such embedding into linear and hyperbolic space incur polynomial and exponential growth, respectively, and are also affected by imbalance in how edges are distributed (i.e. regular graphs are balanced, graphs with hubs are more imbalanced). Some of the difficulty of this problem arises due to dependency in the negative sampling procedure, such that iid assumptions and related results cannot be applied. Towards the end, the authors also show that for the special case of tree data, their theory suggests that the generalization error incurred by a hyperbolic embedding is better than that of a linear embedding. Finally, the authors discuss the limitation that their bound does not depend on the embedding dimension, so is likely not tight. The authors explain their setup through numerous examples of negative sampling strategies, notably the Skip-gram model (ref. [1]), as well as detailed explanations of the different terms appearing in their main result.

**Limitations And Societal Impact:**

The authors have been quite transparent in describing an important limitation of their work (the lack of dependence on embedding dimension). The potentially negative societal impact has not been discussed. I would suggest that the authors change some wording to remove the short, straggling lines throughout the text, thus freeing up space for such a discussion.

**Main Review:**

**Originality**: This paper presents a generalization error bound with interesting consequences and interpretations. Moreover, the proof introduces neat ideas that are of independent interest. Notably, Lemma 3 in the supplementary material gives a nice calculation of the Rademacher complexity of a class of functions defined on data that does not satisfy an iid assumption. The rest of the proof follows a structure similar to that of the main result in ref. [21] (now an ICML paper, so the authors would want to update the reference), so it is nice to compare and contrast the different requirements imposed by the non-iid assumption of the current work.

**Quality**: Continuing from the comments above, I think the paper is quite technically robust. The claims are transparent, and their theoretical justification feels appropriate. The authors use well-placed and judicious remarks to clarify the portions where their approach may fail (e.g. Remark 3 and the Conclusion). The proofs in the supplementary material provide sufficient background, and while I did not verify the details of each inequality, I was able to understand the proof structure at a high level.

**Clarity**: Considering that this is a theory paper, I found the writing quality to be quite nice overall. Once again, the various remarks scattered throughout the text were helpful in guiding the reader.

**Significance:** Given the rapidly growing interest in embedding graph-structured data into hyperbolic space, the results of this paper seem to provide timely guidance in understanding the pros and cons of choosing hyperbolic vs linear space embeddings.

A few small comments:

- Remark 9: "as long as $M$ is large enough" -- could the authors work through a concrete example that practitioners can use in determining the amount of data needed for generalizable hyperbolic embeddings?
- Supplementary, Lemma 3 should read "$\mathcal{X}$ is the domain of functions in $\mathcal{F}$

**Time Spent Reviewing:**

6

---

> ### Author Response · Authors · 2021-08-10
> **Author response**
>
> ### **Comment 1**
> > Remark 9: "as long as $M$ is large enough" -- could the authors work through a concrete example that practitioners can use in determining the amount of data needed for generalizable hyperbolic embeddings?
> $\newcommand{\calX}{\mathcal{X}}$
> $\newcommand{\calF}{\mathcal{F}}$
> $\newcommand{\calV}{\mathcal{V}}$
> $\newcommand{\bbR}{\mathbb{R}}$
> $\newcommand{\bbLD}{\mathbb{L}^{D}}$
> $\newcommand{\frakd}{\mathfrak{d}}$
>
> **Response:** In fact, the concrete condition of $M$ is given in Proposition 2 (line 339, just above Remark 9), whose improved version correcting typos is provided in Section F in the supplementary material. Specifically, the proposition states that if
> $$
> M > \left(\frac{3 \omega_{\bbLD} (R)}{4 |\calV| \mu V_\mathrm{min}^{\bbR^2}} \left(\sqrt{8 \nu_{1, \bbLD} \ln |\calV|} + \sqrt{\ln \frac{2}{\frakd}}\right) + \frac{1}{2 \left(\sqrt{8 \nu_{1, \bbLD} \ln |\calV|} + \sqrt{\ln \frac{2}{\frakd}}\right)} \right)^{2},
> \tag{R1.1}
> $$
> then HOE is better than EOE with probability $1 - \frakd$. Here, $R$ is given in Lemma 1, and $\nu_{1, \bbLD} := \sqrt{(\nu_{1, \bbLD}^{+})^2 + (\nu_{1, \bbLD}^{-})^2}$.
> Nevertheless, the phrase "large enough" is not concrete as you pointed out, we will modify Remark 9 as follows:
>
> - **Remark 9** Proposition 2 implies that if the true dissimilarity is given by the graph distance of a tree, then HGE is better than LGE even if the data is not complete and noisy, if $M$ is larger than the right hand side of (R1.1).
>
> We also give a numerical example in the section Numerical example of Lemma 2 after our response to Comment 2, to help practitioners' understanding, and we will include the discussion in the final version.
>
> ### **Comment 2**
> > Supplementary, Lemma 3 should read "$\calX$ is  the domain of functions in $\calF$."
>
> **Response:** The original phrase that you referred to, "$\calX_{k}$ is the domain of functions $\calX_{k}$" in line 566, included typos indeed and sorry for the typos. To make the presentation succinct and clearer, we will modify the two sentences "Consider a set of functions... the domain of functions in $\calX_{k}$" in lines 565-566 as follows:
>
> "Let $\calX_{1}, \calX_{2}, \dots, \calX_{K}$ be sets. We consider a set $\calF \subset \{(f_{1}, f_{2}, \dots, f_{K}) \mid f_{k}: \calX_{k} \to \mathbb{R}, \textrm{ for $k = 1, 2, \dots, K.$} \}$ of sequences of functions."
>
> $
> \newcommand{\calE}{\mathcal{E}}
> \newcommand{\calV}{\mathcal{V}}
> \newcommand{\calG}{\mathcal{G}}
> \newcommand{\bbR}{\mathbb{R}}
> \newcommand{\bbL}{\mathbb{L}}
> \newcommand{\Rtwo}{\bbR^2}
> \newcommand{\Ltwo}{\bbL^2}
> \newcommand{\calR}{\mathcal{R}}
> \newcommand{\vmin}{V_\mathrm{min}^{\Rtwo}}
> \newcommand{\emax}{\eta_\mathrm{max}}
> \newcommand{\edge}{{(i, j)}}
> \newcommand{\deg}{\mathrm{deg}}
> \newcommand{\frakd}{\mathfrak{d}}
> \newcommand{\mup}{{\mu^+}}
> \newcommand{\mun}{{\mu^-}}
> \newcommand{\rp}{{r^+}}
> \newcommand{\rn}{{r^-}}
> $
>
> ### **Numerical example of Lemma 2**
> We consider the complete balanced $\lambda$-ary ($\lambda \ge 5$) tree with height $h$ ($h > 4$).
> - $|\calV| = \frac{1 - \lambda}{1 - \lambda^h}$,
> - $|\calE| = |\calV| - 1$.
>
> Here, the degree of each vertice other than the root is $\lambda + 1$, we have that
> $\vmin \ge \lambda^{h - 2}$, since the $\lambda^{h - 2}$ vertices whose distance from the root is $h - 2$ has a 6 sub-star which is independent of each other.
>
> If $\lambda = 5$ and $h = 4$,
> - $|\calV| = 156$,
> - $|\calE| = 155$,
> - $\vmin \ge 25$.
>
> As a distribution, we consider the case in Example 1 (a) with $r^+ = r^- = 10^{-4}$ as a specific numerical example.
>
> To apply Lemma 2, we need to calculate $\nu_{1, \Ltwo}$, $R$, and $\mu$.
> We can calculate $\nu_{1, \Ltwo}$ from (79) and (80) and its definition in Proposition 2 in the supplementary material.
> The radius $R$ for $\Ltwo$ to satisfy the minimum risk $\calR^*$ is given by Lemma 1 and [29] (for the derivation, see Appendix of this feedback).
> $\mu$ can be calculated also as in the Appendix of this feedback. As a result, we have $\nu_{1, \Ltwo} = 0.02379...$, $R = 4.3841...$, $\mu = 0.0032$.
>
> According to these calculation results, Lemma 2 states that $M > 1.2 \times 10^7$ is a sufficient condition for HGE to outperform LGE with probability $\frakd = 10^{-1}$. This is the first specific calculation result that **theoretically** guarantees the superiority of HGE to LGE. Nevertheless, tightening the right hand side or deriving a necessary condition could be our future work to strengthen the theory established in this paper.
>
> ### Appendix
> **An upper bound of $R$**
> According to Lemma 1, $\Ltwo$ achieves $\calR^*$. We discuss $R$ to achieve this. As shown in the proof of Lemma 1, it is achieved by embedding with scale $\tau:= \max\\{\frac{3}{4}, \emax\\}$. Here, according to [29]
> $$
> \emax = \max_{\edge \in \calE} \frac{- 2 k \ln \tan \frac{\min_{v = i, j} \frac{2 \pi} {\deg (v) + 1} }{2}}{w_\edge}$$, where $w_\edge$ is the weight of $\edge \in \calE$ and $k$ is the absolute value of the curvature.
> In our case, $w_\edge = 1$, $k = 1$.
> Hence
> $$
> \emax = \max_{\edge \in \calE} - 2 \ln \tan \frac{\min_{v = i, j} \frac{2 \pi} {\deg (v) + 1} }{2}
> $$
> Let $R_{\calG}$ is the radius of the tree $\calG$. If $R \ge R_{\calG} \tau$, then we can achieve $\calR^*$ by $\Ltwo$.
>
> **Definition of $\mu$**:
> Define $U := |\calV| (|\calV| - 1)$ and $\rho := \frac{|\calE|}{U}$.
> $\mu$ can be calculated by $\mu := \min\\{\mu^+, \mu^-\\}$, where $\mu^+ := \frac{1}{U} \frac{1}{(1 - r^+)\rho + r^+} - \frac{1}{U} \frac{r^-}{(1 - r^-) (1 - \rho) + r^-}$ and $\mu^- := \frac{1}{U} \frac{1}{(1 - r^-) (1 - \rho) + r^-} - \frac{1}{U} \frac{r^+}{(1 - r^+)\rho + r^+}$. Here, we had a typo regarding the definition of $\mu$ in the original version indeed and sorry for the typo. The derivation is given as follows.
>
> Each positive pair appears as a positive pair in probability $\frac{1}{U} \frac{1}{(1 - r^+)\rho + r^+}$ and negative pair as $\frac{1}{U} \frac{r^-}{(1 - r^-) (1 - \rho) + r^-}$. Hence, if a positive pair violates a condition (1), we suffer from additional risk of $\mu^+ := \frac{1}{U} \frac{1}{(1 - r^+)\rho + r^+} - \frac{1}{U} \frac{r^-}{(1 - r^-) (1 - \rho) + r^-}$. Likewise, each negative pair appears as a negative pair in probability $\frac{1}{U} \frac{1}{(1 - r^-) (1 - \rho) + r^-}$ and as a positive pair and $\frac{1}{U} \frac{r^+}{(1 - r^+)\rho + r^+}$, and hence if a negative pair violates a condition (1), we suffer from additional risk of $\mu^- := \frac{1}{U} \frac{1}{(1 - r^-) (1 - \rho) + r^-} - \frac{1}{U} \frac{r^+}{(1 - r^+)\rho + r^+}$.
>
> Hence, the minimum achievable risk by $\Rtwo$ is $\calR + \vmin \mu$, where $\mu := \min\\{\mup, \mun\\} = \frac{1}{U} \min\\{\frac{1}{(1 - \rp)\rho + \rp} - \frac{\rn}{(1 - \rn) (1 - \rho) + \rn},  \frac{1}{(1 - \rn) (1 - \rho) + \rn} - \frac{\rp}{(1 - \rp)\rho + \rp}\\}$.

---

> > ### Comment · Reviewer_ixpa · 2021-08-31
> > **Thank you for the comments**
> >
> > Thank you to the authors for the additional examples. After going through the other reviews and responses, my opinion is that this is a good paper. However, there is a slight caveat with the readability that other reviewers have brought up, and I hope that the authors will take their comments into account during revisions. I will maintain my score of 7.

---

> > > ### Author Response · Authors · 2021-09-03
> > > **Thank you for the reply.**
> > >
> > > We appreciate your comments thinking highly of our work. We will further improve the readability of the final version by reflecting on your feedback.

---

### Decision · Program_Chairs · 2021-09-27

**Decision:**

Accept (Poster)

**Comment:**

The paper considers generalization error incurred when embedding graphs in linear versus hyperbolic space, in particular in the context of embeddings obtained via negative sampling, and presenting results where linear/hyperbolic space incurs polynomial/exponential growth, respectively, and well as on edge heterogeneity.  One low reviewer raised his/her score in the discussion phase, and
still has the good suggestion to incorporate comments to improve readability.